# Repeated origins, widespread gene flow, and allelic interactions of target-site herbicide resistance mutations

Julia M Kreiner[1]*‡§, George Sandler[1], Aaron J Stern[2], Patrick J Tranel[3], Detlef Weigel[4], John R Stinchcombe[1]†, Stephen I Wright[1]†

[1]Department of Ecology and Evolutionary Biology, University of Toronto, Toronto, Canada; [2]Graduate Group in Computational Biology, University of California, Berkeley, Berkeley, United States; [3]Department of Crop Sciences, University of Illinois Urbana-Champaign, Urbana, United States; [4]Department of Molecular Biology, Max Planck Institute for Biology Tübingen, Tübingen, Germany

**\*For correspondence:**
julia.kreiner@ubc.ca

†These authors contributed equally to this work

**Present address:** ‡Department of Botany, University of British Columbia, Vancouver, Canada; §Biodiversity Research Centre, University of British Columbia, Vancouver, Canada

**Abstract** Causal mutations and their frequency in agricultural fields are well-characterized for herbicide resistance. However, we still lack understanding of their evolutionary history: the extent of parallelism in the origins of target-site resistance (TSR), how long these mutations persist, how quickly they spread, and allelic interactions that mediate their selective advantage. We addressed these questions with genomic data from 19 agricultural populations of common waterhemp (*Amaranthus tuberculatus*), which we show to have undergone a massive expansion over the past century, with a contemporary effective population size estimate of $8 \times 10^7$. We found variation at seven characterized TSR loci, two of which had multiple amino acid substitutions, and three of which were common. These three common resistance variants show extreme parallelism in their mutational origins, with gene flow having shaped their distribution across the landscape. Allele age estimates supported a strong role of adaptation from de novo mutations, with a median age of 30 suggesting that most resistance alleles arose soon after the onset of herbicide use. However, resistant lineages varied in both their age and evidence for selection over two different timescales, implying considerable heterogeneity in the forces that govern their persistence. Two such forces are intra- and inter-locus allelic interactions; we report a signal of extended haplotype competition between two common TSR alleles, and extreme linkage with genome-wide alleles with known functions in resistance adaptation. Together, this work reveals a remarkable example of spatial parallel evolution in a metapopulation, with important implications for the management of herbicide resistance.

## Editor's evaluation

This paper studies the evolution of herbicide resistance in *Amaranthus tuberculatus*, a widespread agricultural weed. By illuminating how adaptive mutations arose and spread in this remarkable example of rapid human-induced adaptation, the study will be of interest to a broad audience, ranging from plant biologists interested in herbicide resistance to evolutionary biologists and population geneticists studying the fundamental factors and processes that govern rapid adaptation. The paper applies innovative population genetic methodology to support its primary finding that resistance mutations have evolved multiple times in parallel.

## Introduction

The evolution of resistance in agricultural pest populations occurs rapidly and repeatedly in response to herbicide and pesticide applications. Reports of herbicide resistance across agricultural landscapes have been steadily growing, threatening crop productivity and greatly raising costs for agricultural production (*Peterson et al., 2018*). These reports put a lower limit on the estimated number of unique resistance cases—over 500 across the globe—based on just the occurrence of resistance to different herbicide modes-of-action across different species (*Heap, 2014*) and barring the probably minor role of interspecific hybridization. For acetolactate synthase (ALS) inhibiting herbicides alone, over 160 species have evolved resistance since the first report in 1986, which was only five years after their initial introduction (*Comai and Stalker, 1986*; *Heap, 2014*; *Whitcomb, 1999*). These numbers are likely a vast underestimate of the repeatability of herbicide resistance evolution. For ALS herbicides, for example, non-synonymous substitutions at eight distinct codons confer resistance, with most of them found in multiple species (*Tranel and Wright, 2002*), and with multiple independent causal mutations often occurring in the same population (*Heap, 2014*; *Kreiner et al., 2018*). In addition to repeated resistance evolution through distinct causal resistance loci, it is likely that for a single locus, resistance mutations have arisen repeatedly within a species (*Kreiner et al., 2019*). While these observations suggest herbicide resistance may be among the most extreme cases of contemporary parallel evolution in plants, it remains unclear how often resistance is spread across the range through gene flow versus repeated independent origins.

Population genomic approaches can greatly help to understand the origin and spread of herbicide resistance. Genomic methods have tested for differences in population structure among resistant and susceptible agricultural populations (*Küpper et al., 2018*), reconstructed complex genomic regions associated with resistance (*Molin et al., 2017*), and investigated patterns of selection on and the extent of convergence between loci conferring non-target site resistance (*Kreiner et al., 2020*; *Van Etten et al., 2019*). But even for validated resistance mutations that occur within the gene whose product is targeted by the herbicide (target-site resistance [TSR] mutations), investigations of their recent evolutionary history are sparse (but see *Flood et al., 2016*; *Kreiner et al., 2019*). With large-effect mutations identified as being causal for conferring target-site resistance to nine herbicides at 19 loci across many species (*Murphy and Tranel, 2019*), the field is ripe for the application of population genomic techniques for resolving the evolutionary history of herbicide resistance and informing integrative management strategies in weed populations.

In contrast to most of the selective sweep literature coming from within-host studies of drug resistance in HIV (e.g. *Feder et al., 2016*; *Pennings et al., 2014*)—where sweeps occur in a closed-system, often starting from a single founding viral lineage and evolving within individual patients—evolutionary patterns of resistance to herbicides across a relevant agricultural landscape are by no means expected to be as tidy (but see *Feder et al., 2017*; *Feder et al., 2019* for spatial structure in HIV evolution). Weedy agricultural populations themselves, or at least genotype compositions, may be transient in space and time due to widespread gene flow and changing selection regimes through rotations of both focal crops and herbicide applications (*Holst et al., 2007*; *Naylor, 2003*; *Neve et al., 2009*). Consequently, persistent agricultural weed populations likely comprise a collection of resistant haplotypes that have arisen and dispersed across the landscape, following a model of spatial parallel mutation in an interconnected network of populations (*Ralph and Coop, 2010*). Recent population genomic evidence supports this prediction for a subset of newly problematic glyphosate-resistant agricultural populations of common waterhemp (*Amaranthus tuberculatus*) in Ontario, Canada, where both multiple origins and long-distance dispersal contributed to the spread of glyphosate resistance (*Kreiner et al., 2019*).

*Amaranthus tuberculatus* is a major challenge for agricultural practices in the Midwestern US, and is among the most problematic weeds worldwide in terms of its capacity for evolving resistance to multiple herbicides (*Tranel, 2021*). Conforming to classic hypotheses about successful weeds (*Baker, 1974*), the species has large census population sizes, a widespread distribution, and considerable seedbanks (*Costea et al., 2005*). The dioecious, wind-pollinated *A. tuberculatus* additionally offers an obligately outcrossing mating system, providing more independent backgrounds on which new mutations can arise (*Costea et al., 2005*; *Kreiner et al., 2018*) and an effective dispersal system (*Liu et al., 2017*). The species has not always been troublesome—the plant is native to North America, where it likely has grown in riparian habitats long before the advent of modern agriculture (*Sauer,*

*1957*). While it persists in these habitats, in the past 100 years or so, *A. tuberculatus* has become strongly associated with agricultural fields (*Costea et al., 2005*; *Tranel and Trucco, 2009*). Over the past three decades, *A. tuberculatus* has evolved resistance to seven chemical modes-of-action (*Heap, 2021*; *Tranel, 2021*), including resistance to both ALS-inhibiting herbicides and protoporphyrinogen oxidase (PPO)-inhibiting herbicides (*Shoup et al., 2003*).

ALS-inhibiting herbicides have been among the most popular mode-of-action for weed control in crops since their introduction in the 1980s (*Brown, 1990*), and are widely used in both corn and soy production systems. They were rapidly adopted due to their application rates being an order of magnitude lower than previous herbicides and thus increased affordability, along with low toxicity and broad-spectrum weed control (*Mazur and Falco, 1989*), but quickly became notorious for their ability to select for resistant weeds (*Tranel and Wright, 2002*). Use of ALS herbicides thus decreased in the 1990s, coinciding with the widespread adoption of Round-up ready cropping systems, of which glyphosate herbicides are an essential component (*Green, 2007*). While historically not as popular as ALS-inhibiting herbicides, PPO-inhibiting herbicides have been used for nearly 50 years for the control of dicotyledonous (broadleaf) weeds, at its peak in the early 1990s representing 10% of annual applications in the USA but dropping to 1.5% by 2006 (U.S. Department of Agriculture, National Agricultural Statistics Service (USDA-NASS), 2012). However, PPO-inhibiting herbicides have since seen a resurgence for control of weeds that have evolved resistance to heavily-used herbicides such as glyphosate and ALS inhibitors (*Dayan et al., 2018*; *Tranel, 2021*; *Tranel and Wright, 2002*; *Zhao et al., 2020*).

Here, we investigate the evolutionary histories of mutations that have been previously demonstrated to confer target-site resistance in *A. tuberculatus*, focusing on TSR mutations within ALS and PPO. We infer the number of TSR mutational origins across populations and their distribution across the landscape, examining the signals left behind by both mutation and recombination. Specifically, we implement a method that infers the ancestral recombination graph (ARG) and that offers a powerful approach for inference of selective history, by providing near-complete information on relatedness among haplotypes (*Rasmussen et al., 2014*). Coupled with estimates of effective population size ($N_e$) through time based on coalescent rates across the genome (*Speidel et al., 2019*), these methods allow for powerful hypothesis testing on the role of standing variation versus new mutation. We assess heterogeneity in whether independent resistant lineages are associated with pronounced signals of selection based on a tree-based test and selective sweep signals, some of which may be mediated by intra- and inter-locus allelic interactions. We also examine signatures of the haplotype competition between common ALS resistance alleles, and the extent that extreme selection from herbicides on TSR mutations has impacted diversity across the genome. Our detailed population genomic analysis describing the repeatability of and heterogeneity in target-site herbicide resistance evolution advances our understanding of rapid adaptation of multicellular organisms to extreme selective pressure, while providing evolutionary informed priorities for agricultural weed management.

## Results
### Types of target-site mutations

To test hypotheses about the origins of TSR in *Amaranthus tuberculatus*, we used whole-genome sequence information from 19 agricultural fields in the Midwestern US and Southwestern Ontario, Canada. It is important to note that these populations were obtained from fields where *A. tuberculatus* was only poorly controlled, potentially overrepresenting the frequency of resistance across the landscape.

Having previously characterized two types of target-site glyphosate resistance in these samples (coding sequence substitutions and gene amplification; *Kreiner et al., 2019*), here we focus on all other characterized mutations known to confer resistance in the genus Amaranthus. We examined our sequence data for the presence of eight such substitutions in ALS, three in PPO, and one in photosystem II protein D1 (psbA - the target of a class of herbicides that inhibits electron transfer). Across 152 individuals, we found segregating variation at six out of eight known ALS mutations, and one of the three known PPO mutations (*Table 1*). We did not find any mutation in psbA.

**Table 1.** Number and frequency of resistant individuals and mutations at loci known to be causal for resistance to PPO and ALS herbicides, both totals, and within each agricultural region. Relative frequencies given in parentheses.

| | PPO | ALS | | | | | | | |
| --- | --- | --- | --- | --- | --- | --- | --- | --- | --- |
| | ΔGly210 | Trp-574-Leu | Ser-653-Asn | Ser-653-Thr | Gly-654-Phe | Pro-197-Leu | Pro-197-His | Ala-122-Ser | Asp-376-Glu |
| Total number of individuals with TSR mutations | 22 (0.145) | 80 (0.526) | 48 (0.316) | 2 (0.013) | 1 (0.007) | 1 (0.007) | 1 (0.007) | 2 (0.013) | 3 (0.020) |
| Total number of TSR mutations | 25 (0.082) | 106 (0.349) | 59 (0.194) | 2 (0.007) | 1 (0.003) | 1 (0.003) | 1 (0.003) | 2 (0.007) | 3 (0.010) |
| TSR mutations in Walpole, Ontario, Canada | 0 | 12 (0.162) | 12 (0.162) | 1 (0.0135) | 0 | 0 | 0 | 0 | 0 |
| TSR mutations in Essex, Ontario, Canada | 1 (0.0125) | 23 (0.288) | 35 (0.438) | 0 | 1 (0.0125) | 0 | 1 (0.0125) | 2 (0.025) | 3 (0.0375) |
| TSR mutations in Midwestern US | 24 (0.16) | 71 (0.47) | 12 (0.08) | 0 | 0 | 1 (0.0067) | 0 | 0 | 0 |

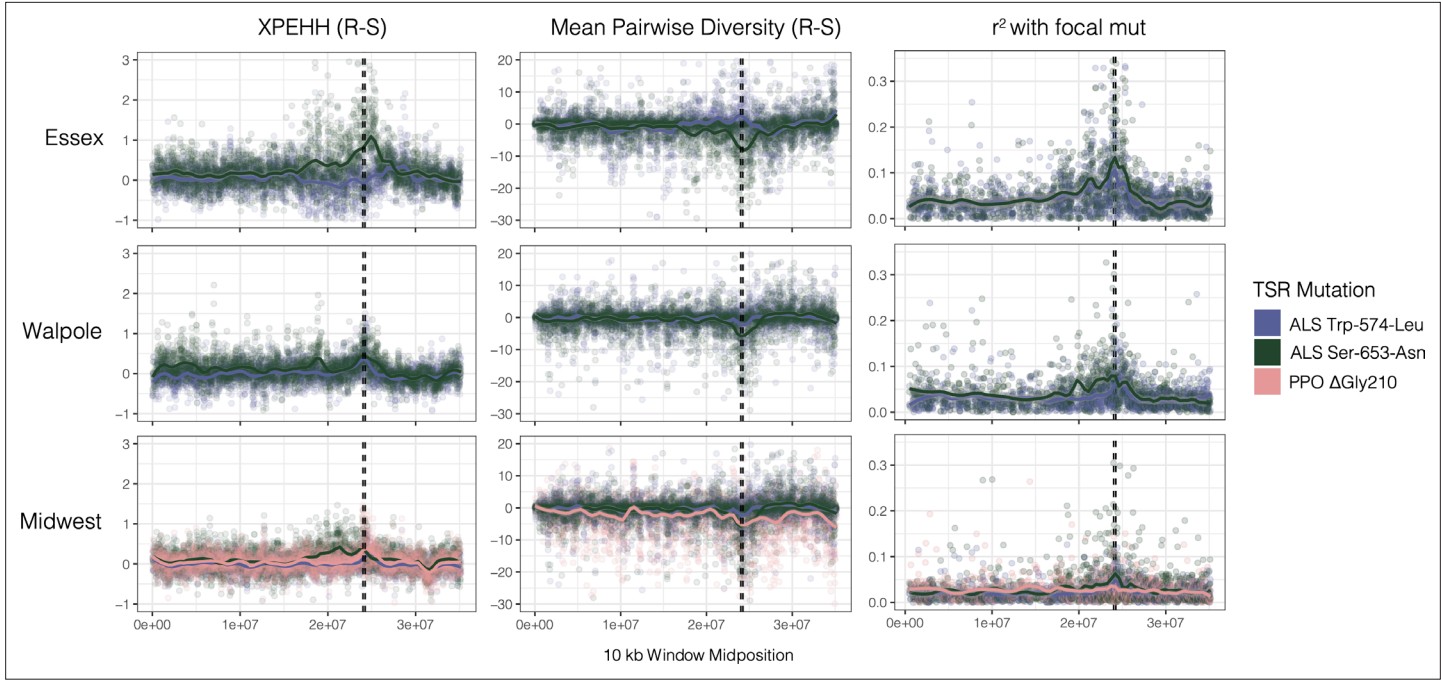

**Figure 1.** Sweep-scan summary statistics by geographic region. (Left) Difference in integrated haplotype homozygosities (XPEHH) between haplotypes carrying the focal TSR mutation and susceptible haplotypes. (Middle) Difference in mean pairwise diversity between haplotypes carrying the focal TSR mutation and susceptible haplotypes. (Right) $r^2$ of other missense mutations with focal TSR mutation on genotype, rather than haplotype data. In all columns, dashed vertical lines denote PPO (left) and ALS (right) genes, which are only 250kb apart in the genome.

The nine unique PPO and ALS target-site resistant mutations occur at seven distinct codons, with two positions segregating for multiallelic resistance: two non-synonymous changes at codons 197 and 653 in the ALS gene. Six out of nine variants are rare, with five or fewer instances, in contrast to the common Trp-574-Leu and Ser-653-Asn nonsynonymous substitutions in ALS, and the PPO ΔGly210 deletion (*Table 1*). Notably, the most common resistance mutation (referring to identity-by-state), Trp-574-Leu, is found in 53% of agricultural individuals, and the second most common, Ser-653-Asn, in 32% of individuals (*Table 1*). From these two most frequent ALS mutations alone, 74% of individuals sampled had resistance to ALS-inhibiting herbicides. Accounting for rare ALS resistance mutations only increases this percentage to 75%, because these rare variants are almost exclusively found in individuals already harboring one of the two common ALS mutations.

Regardless of the geographic region (within Essex County, Walpole Island, and the Midwestern US), multiple causal changes confer ALS resistance. Furthermore, the majority of populations (5/8 populations within the Midwestern US, 5/5 populations in Essex County, and 4/6 populations in Walpole) harbor multiple causal ALS mutations (*Table 1*). Thus, at just the level of these distinct mutational types, we observe genetic convergence in adaptation to ALS-inhibiting herbicides at global, regional, and population scales.

## Regional selective sweep signals

To learn how and how often the individual mutations might have arisen, we first visualized regional selective sweep patterns at PPO and ALS genes—two genes that are located only ~250 kb apart in the genome—with respect to the common Trp-574-Leu, Ser-653-Asn, ΔGly210 alleles. In particular, we assayed the extent to which selection from herbicides at these genes has led to reductions in diversity, and increases in homozygosity and linkage across the haplotype, as would be expected if TSR alleles have increased in frequency rapidly enough that recombination has yet to unlink these alleles from the background on which they arose. We found that corresponding selective sweep signals appear to be highly heterogeneous across geographic regions and across resistance mutations (*Figure 1*). The most pronounced selective sweep signal at the regional level is for the ALS Ser-653-Asn mutation, in our large collection of nearby populations from Essex County. These resistant haplotypes show a dramatic

excess of homozygosity over susceptible haplotypes for nearly 10 Mb (XPEHH, *Figure 1* top-left green line). The breadth of the impact of selection on local chromosome-wide linkage disequilibrium (LD) is worth noting—this extended sweep signal is even larger than what was seen for an EPSPS-related gene amplification whose selective sweep in response to glyphosate herbicides spanned 6.5 Mb in these same individuals (*Kreiner et al., 2019*). This pattern of a hard selective sweep is also apparent in patterns of pairwise diversity and in LD of the focal Ser-653-Asn mutation with missense SNPs ($r^2$) (*Figure 1*, top middle/right). In contrast, the Trp-574-Leu mutation in Essex actually shows a slight excess of heterozygosity and excess diversity compared to susceptible haplotypes, but nearly as strong LD with other missense SNPs (*Figure 1*, purple line top row).

Selective sweep signals are much subtler in Walpole and especially in the Midwestern US compared to Essex. In Walpole as in Essex, ALS resistance shows a stronger signal of selection for Ser-653-Asn than for Trp-574-Leu, whereas signatures of selection for either mutation are almost completely lacking from the Midwestern US except for a slight peak in $r^2$ for ALS Ser-653-Asn. The PPO ΔGly210 mutation is found at considerable frequencies only in the Midwestern US, but regional sweep signals based on homozygosity, diversity, and LD are absent with respect to the deletion (*Figure 1*, pink line bottom row).

Despite inconsistent sweep signals, the mutations we describe here are extremely likely to have experienced selection over their history, but varying over space and time. We know from previous functional validation that these mutations are causal for resistance to ALS or PPO inhibiting herbicides [in *Amaranthus tuberculatus* for the PPO deletion, as well as ALS Trp-574-Leu, and both Ser-653-Asn and Ser-653-Thr substitutions (*Foes et al., 2017*; *Patzoldt and Tranel, 2017*; *Shoup et al., 2003*), and in congeners for the remaining mutations (*McNaughton et al., 2001*; *Nakka et al., 2017*; *Singh et al., 2018*; *Whaley et al., 2004*)]. Thus, we set out to identify the extent to which repeated origins and gene flow have influenced regional signatures of selection, as well as identify key processes that may underlie heterogeneity in their recent evolutionary histories.

## Inferring the genealogical history of target-site resistance mutations

We first took a gene tree approach to reconstruct the evolutionary history of TSR mutations, based on phased haplotypes inferred from performing the most up-to-date joint population and read-backed phasing methods (SHAPEIT4 *Delaneau et al., 2019*; WhatsHap v1.0 *Martin et al., 2016*). We found that patterns of similarity among phased haplotypes at ALS and PPO (including 5 kb upstream and downstream of either gene) indicated numerous origins for every common resistance mutation: PPO ΔGly210, ALS Trp-574-Leu, and Ser-653-Asn (*Figure 2—figure supplement 1*). A gene tree based on raw pairwise differences between haplotypes, as illustrated here, sets an upper limit on the number of independent origins for each mutation. Because recombination causes ancestral haplotypes to decay in size as they are passed down through time, linked sites may not necessarily have identical genealogies as a single mutational origin may be recombined onto distinct haplotypes. In cases such as this, ancestral recombination graphs (ARGs) can allow for more accurate inferences of genealogical history by generalizing the inference of coalescent history along a recombining unit (*Griffiths and Marjoram, 1997*; *Griffiths and Marjoram, 1996*; *Hudson, 1983*).

We reconstructed the ARG for 20,000 SNPs encompassing both ALS and PPO genes (a ~ 1 and ~ 10 kb gene, respectively, separated by 250 kb on the same chromosome) using ARGweaver (*Hubisz and Siepel, 2020*; *Rasmussen et al., 2014*). We assessed the likelihood of the ARG inferences under varying constant recombination rates and over two time step parameters (*Figure 2—figure supplements 2 and 3*). From our most likely parameter values (recombination = $10^{-8}$, time steps = 30), and based on the MCMC sample that maximizes the likelihood of our data across 1,250 iterations, we extracted the tree corresponding to each focal TSR locus. For all three common TSR mutations, ALS Trp-574-Leu, ALS Ser-653-Asn, and PPO ΔGly210, we found evidence for multiple independent origins producing the same resistant variant—three for ALS Trp-574-Leu, two for ALS Ser-653-Asn, and two for PPO ΔGly210 (*Figure 2A*). Support for these origins was generally very high, with 5/7 origins showing that 100% of MCMC samples were consistent with each cluster of haplotypes being monophyletic. The two origins with less than full support were haplotypes harbouring the ALS Trp-574-Leu, where one high-frequency origin had 85% support (105/125 MCMC samples) and a low-frequency origin had 45% support (56/125), implying that occasionally haplotypes mapping to these origins belonged to other groupings across MCMC samples. The findings of multiple origins of identical resistance

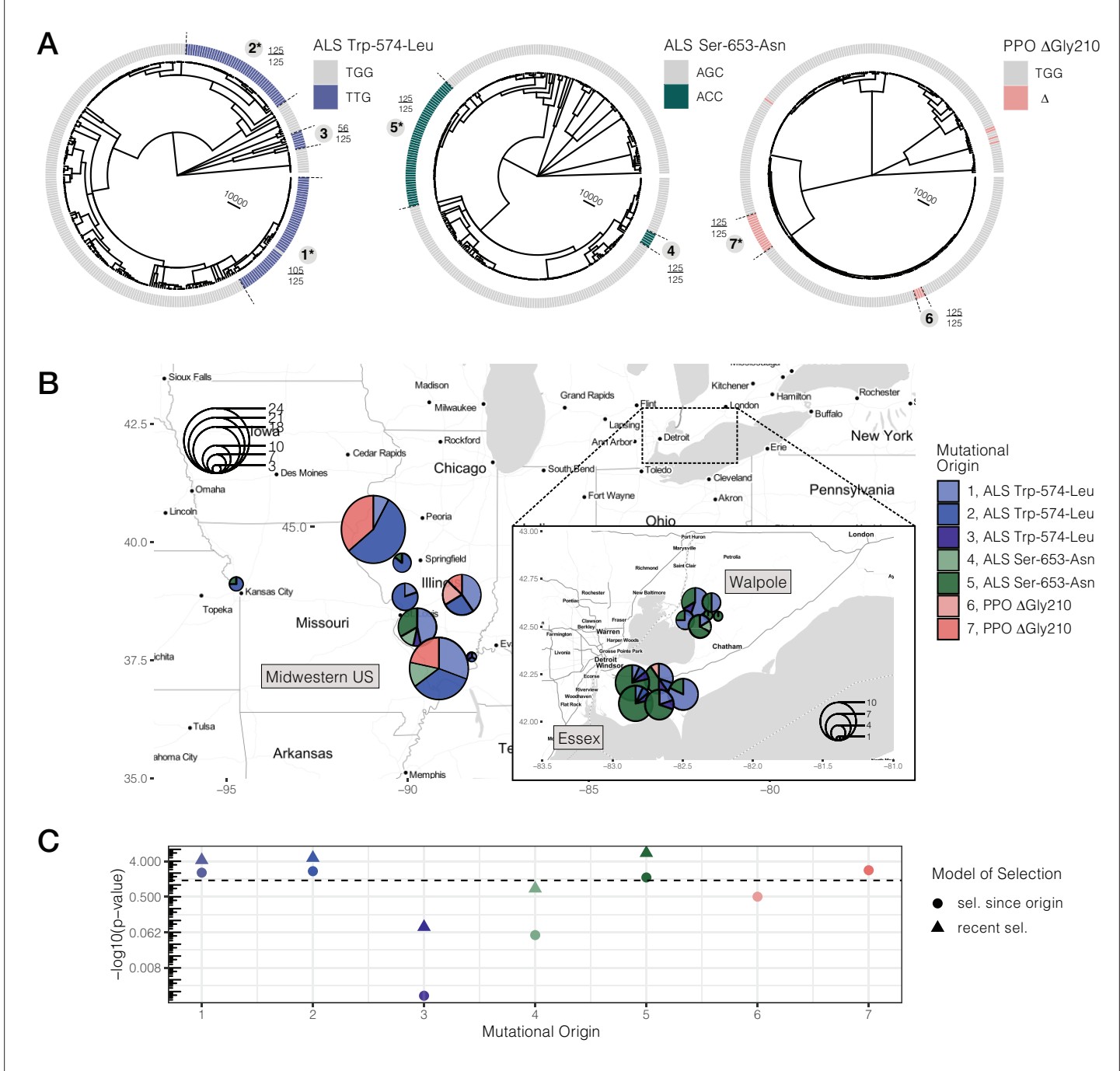

**Figure 2.** Repeated-independent origins and range-wide distribution of three target-site resistance mutations, along with their associated significance of selection over two different timescales. (**A**) Trees at focal TSR loci corresponding to an ARG estimated across 20 kb SNPs. Bold numbers around trees identify clusters of resistant haplotypes consistent with independent origins. The presence of an asterisk at each origin number implies significant evidence of selection since the mutation arose de novo at p < 0.05 against the null distribution, as in C. Support for monophyly for each origin across 125 samples of 1250 MCMC iterations is depicted by the fraction found outside each cluster. (**B**) Geographic distribution of haplotypes originating from distinct mutational lineages as inferred from A. TSR mutational lineages are found across numerous populations and agricultural regions, although regions show clear differences in the frequency of some mutations. (**C**) Results of tree-based tests of non-neutral allele frequency change (*Speidel et al., 2019*) from each mutational origin of TSR under two alternative models of selection; selection on a mutation since its origin versus selection over more recent timescales (on the last 0.01% of the tree). The horizontal dashed line denotes the p-value cutoff of α = 0.05, after false discovery rate correction.

The online version of this article includes the following source data and figure supplement(s) for figure 2:

**Source data 1.** Tree-based coalescent test for selection under two scenarios; selection on since the mutation first arose and selection even more recent

*Figure 2 continued on next page*

*Figure 2 continued*

timescales (i.e.the last 1% of the tree).

**Source data 2.** Tree sequence corresponding to ALS Trp-574-Leu, ALS Ser-653-Asn, and PPO ΔGly210 extracted from the most likely iteration of the ARGweaver MCMC.

**Source data 3.** Resistance status at ALS Trp-574-Leu, ALS Ser-653-Asn, and PPO ΔGly210 for haplotypes mapped in *Figure 2*.

**Figure supplement 1.** Bootstrapped gene trees of ALS (3 kb) and PPO (10 kb) (coding sequence + 1 kb on either side) alongside TSR mutations across all 162 individuals.

**Figure supplement 2.** The influence of both the number of timesteps coalescent events are estimated over (t) and constant recombination rate magnitude (r) on ARG likelihood across 1250 MCMC iterations.

**Figure supplement 3.** At the most likely number of (timesteps = 30), the influence of increasing the recombination rate parameter constant value from $r = e^{-7}$ to $r = e^{-9}$ on the tree sequence inference from the most likely ARG.

**Figure supplement 4.** Phased haplotypes corresponding to distinct origins of target-site resistance mutations at the ALS Trp-574-Leu (Left side, grey vertical dashed line), ALS Ser-653-Asn (left side, black vertical dashed line) and PPO ΔGly210 (right side, dashed vertical white line) positions, relative to susceptible lineages.

**Figure supplement 5.** XPEHH, cross-population extended haplotype homozygosity, for haplotypes mapping to each origin of ALS Trp-574-Leu, ALS Ser-653-Asn, and PPO ΔGly210 (1–7, top to bottom, plus unplaced PPO ΔGly210 haplotypes as the last row) in comparison to susceptible haplotypes.

**Figure supplement 6.** H12, homozygosity of the two most common haplotypes, for each origin of ALS Trp-574-Leu, ALS Ser-653-Asn, and PPO ΔGly210 (1–7, top to bottom, plus unplaced PPO ΔGly210 haplotypes as the last row).

mutations, and most specific origin scenarios, are consistent even across less likely recombination rate parameterizations (*Figure 2—figure supplements 2 and 3*). Compared to the gene trees based on the average mutational history across the PPO and ALS (*Figure 2—figure supplements 1 and 4*), accounting for recombination has clearly further resolved the origins of these target-site resistance alleles. In comparison to susceptible haplotypes, resistant haplotypes at each origin are apparently more homozygous but retain some diversity, presumably driven by both mutation and recombination subsequent to their origin (*Figure 2—figure supplements 4–6*).

When haplotypes belonging to distinct mutational lineages are mapped across populations (*Figure 2B*), it is clear that, despite the many independent origins, gene flow has also played a major role in the spread of resistance across the landscape. Haplotypes from the three most common origins of resistance to ALS herbicides —Trp-574-Leu #1, #2, and Ser-653-Asn #5 (corresponding to 39, 25, and 47 haplotypes, respectively)—were identified in 15, 10, and 14 populations. To test if mutational lineages were more geographically structured than expected given their frequency, we performed a permutation of haplotype assignment to a geographic region. All but two mutational lineages were consistent with expectations under this null, suggesting near panmixia between the Midwestern US and Ontario (Essex + Walpole) for most resistant lineages. The exceptions were ALS Trp-574-Leu #2, which is exclusive to the Midwestern US (44/47 resistant haplotypes mapping to the Midwest or 94%; 95% CIs of regional permutations [0.383,0.638]), and Ser-653-Asn #5, which is at a much higher frequency in Ontario populations (87% found in either Essex or Walpole; 95% CIs of regional permutation [0.370, 0.630]), suggesting that these mutations arose locally in these geographic regions and have not yet had the chance to spread extensively.

We next performed a tree-based test of non-neutral allele frequency change to examine whether TSR alleles have experienced consistent or shifting selection over their histories. Specifically, we implemented a tree-based statistic that relies on the order of coalescent events (*Speidel et al., 2019*), in addition to a modified version of this statistic that evaluates the probability of selection on more recent timescales (Materials and methods, *Tree-based tests for selection*). We approached these tests of selection one unique mutational origin at a time, excluding all other resistant lineages from the tree, such that our estimates of the probability of selection for a given mutational origin are relative to all other susceptible lineages.

Under the scenario of consistent selection since the origin of the mutation, four out of seven mutational origins we tested were significant at α = 0.05 after a 5% false discovery rate (FDR) correction (*Figure 2C*, *Figure 2—source data 1*). Since it arose, the Midwestern US high-frequency ALS Trp-574-Leu #2 variant showed the strongest signature of selection across all origins and all resistant loci (p = 0.0058), followed by the widespread high-frequency ALS Trp-574-Leu #1 variant (p = 0.0089),

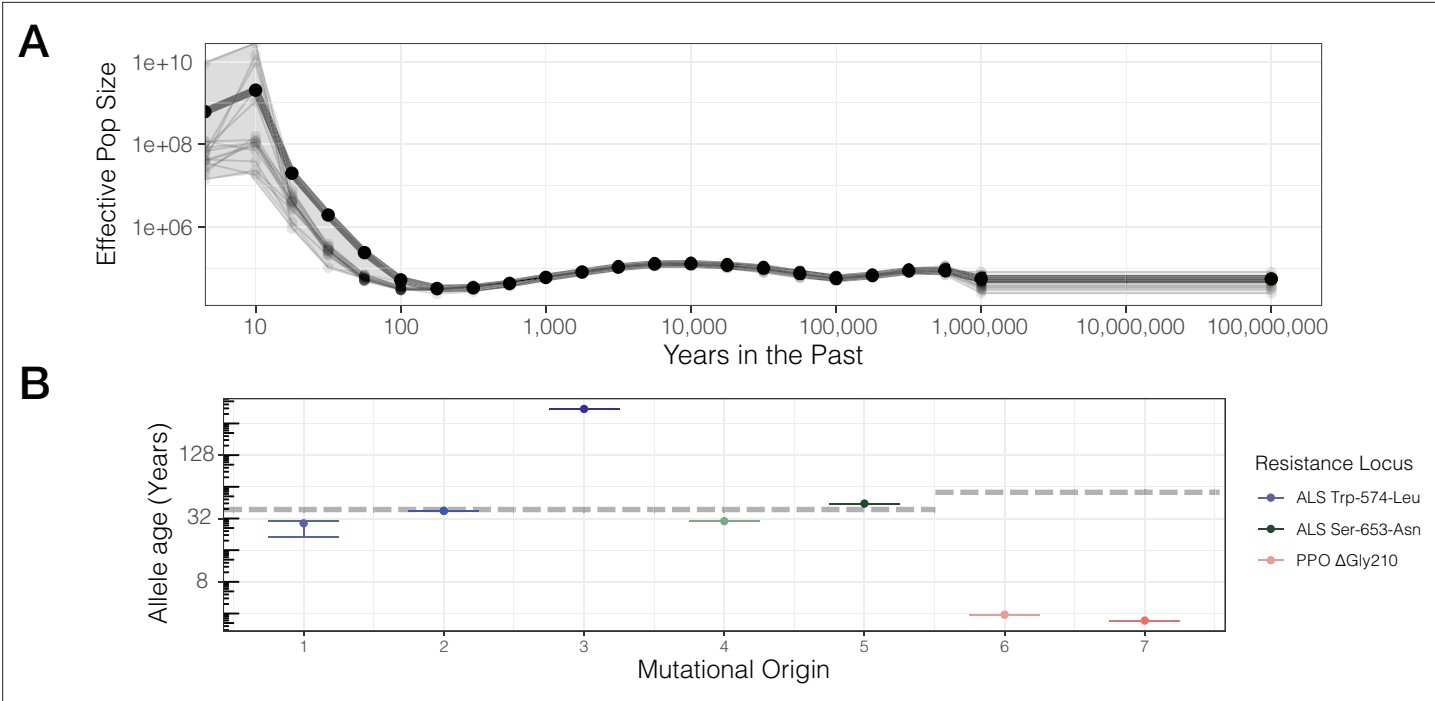

**Figure 3.** Contemporary population expansion of *A. tuberculatus* and corresponding ages of TSR variants. (**A**) Relate-inferred effective population size through time, illustrating a remarkable population expansion occurring over the last 100 years. The bold line indicates results from genome-wide SNPs, whereas thinner lines represent results from chromosome-by-chromosome analyses, with the shaded area showing the bounds of the variance in the chromosome-by-chromosome data. (**B**) Allele age inferred from the geometric mean effective population size estimate over the timescale of contemporary herbicide use ( < 50 years ago, $GM[N_e]$ = 83,294,700). Horizontal dashed lines for ALS Trp-574-Leu and ALS Ser-653-Asn, and PPO origins represent the approximate onset of ALS and PPO herbicide use, respectively.

The online version of this article includes the following source data for figure 3:

**Source data 1.** Effective population size estimates for each chromosome, and genome-wide, from Relate.

**Source data 2.** Unscaled allele age estimates corresponding to the seven inferred origins of ALS Trp-574-Leu, ALS Ser-653-Asn, and PPO ΔGly210.

the common Midwestern US PPO variant #7 (p = 0.00434), and lastly, the high-frequency Ontario ALS Ser-653-Asn #5 variant (p = 0.02831).

In addition, a test for selection over the most recent 1% of the tree showed that for three of the five selected lineages that predated this time cutoff, there is even stronger evidence for selection on more recent timescales, after FDR correction (***Figure 2C***, ***Figure 2—source data 1***). The most obvious of these is the ALS Ser-653-Asn variant #5, which, while having the weakest evidence of consistent selection over its history, shows the strongest evidence of selection over recent timescales (p = 3.44 × 10⁻⁷). These tests therefore illustrate a strong role for fluctuating selection, intensifying over recent timescales.

While our previous test provides insight into the consistency of selection across the course of a mutational lineage's history, a conceptually related approach is to directly assess the role of resistance adaptation from standing genetic variation or new mutation based on allele age estimates relative to the onset of the selection. Allele age estimates depend greatly on the accuracy of effective population size estimates over the relevant evolutionary timescale. Namely, for herbicide resistance evolution, we posit that the relevant $N_e$ is most likely the effective population size over the last half-century or less, corresponding with the introduction of agronomic pesticide regimes. While we have previously used δaδi (***Gutenkunst et al., 2009***) to model species-wide demography in a two-epoch model and found a large population size expansion (historical epoch $N_e$ ~500,000; recent epoch var. *rudis* $N_e$ ~5,000,000; ***Kreiner et al., 2019***), we now used Relate to infer effective population size through time from genome-wide tree sequence data on even more recent timescales (***Speidel et al., 2021***; ***Speidel et al., 2019***). Historical $N_e$ between 100 and 1,000,000 years ago appears to have stayed relatively consistent, with a harmonic mean of ~63,000 (standard error across chromosomes ± 7000 years)

(*Figure 3A*). However, our samples show evidence for massive recent population expansion over the last 100 years, with the contemporary geometric mean $N_e$ estimate 3–4 orders of magnitude higher than the historical $N_e$ (*Figure 3A*)—80,000,000 over the timescale of ALS herbicide use (approximately the last 40 years).

Based on our contemporary $N_e$ estimates relevant to the timescale of herbicide use, we rescaled allelic ages for distinct mutational origins across our ARG-inferred trees, accounting for variation across 1250 MCMC ancestral recombination graph iterations. We found considerable variation in the estimated age of resistance mutations across distinct lineages of the same mutation and across the three different TSR loci, according to haplotype groupings from the most likely ARG (*Figure 3B*). Across all mutational lineages, even for those with slightly lower support for monophyly (ALS Trp-574-Leu #1, #3), the 95% confidence interval of allele ages show these estimates are highly consistent across MCMC sampling of the ARG (*Figure 3B*). Our estimates of the age of the PPO ΔGly210 lineages are extremely recent and much less variable compared to ALS Trp-574-Leu and ALS Ser-653-Asn lineages (median PPO ΔGly210 age = 3.7 years [SE = 0.2]; ALS Ser-653-Asn = 37.8 [SE = 7.1]; ALS Trp-574-Leu = 37.2 [SE = 106.3]). With ALS-inhibiting herbicides having been introduced in the early 1980s, only one lineage (ALS Trp-574-Leu #3) appears to substantially predate herbicide, with an estimated age of ~350 years. However, this lineage is also the one with the least support for monophyly, which will upwardly bias estimates of allele age. Thus, while the exact timescales highly depend on accurate estimation of contemporary effective population sizes, the results are generally consistent with most mutations arising very recently after the onset of herbicide use.

In aggregate, our analyses have uncovered multiple independent origins of large-effect resistance mutations, along with heterogeneity in their evolutionary histories, from the timescale over which they have persisted to their associated signatures of selection. The spread of these parallel origins across the landscape further allows us to observe how these alleles interact when they meet (*Ralph and Coop, 2010*), and how this interaction may be modified by other alleles across the genome.

## Haplotype competition and inter-locus interactions of target-site resistance mutations

While 16 individuals harbour both the common ALS Trp-574-Leu and ALS Ser-653-Asn mutations, haplotype-level analyses indicate that no single haplotype harbours both mutations (*Figure 2—figure supplements 1 and 4*). This lack of double resistant haplotypes is a strong violation of expectations under linkage equilibrium ($\chi^2_{df=1}$ = 16.18, p = 5.77 × 10$^{-5}$), further suggesting that no or very little recombination has occurred between these sites. Given how globally common these resistance alleles are (*Table 1*; 53% and 32% of individuals harbour ALS Trp-574-Leu or Ser-653-Asn), their coexistence, yet independence, suggests that allelic competition may be important in adaptation to ALS-inhibiting herbicides.

Locally in Essex, where haplotypes carrying either the ALS Trp-574-Leu or ALS Ser-653-Asn mutations segregate at intermediate frequencies (29% and 44% respectively; *Table 1*), signed linkage disequilibrium (LD) between genotypes at these two sites is considerably negative ($r$ = –0.67). In comparison to other non-synonymous SNPs at similar frequencies (minor allele frequency > 0.20) and separated by similar distances (≤ 500 bp) across the genome, this level of repulsion is unexpected (one-tailed p = 0.033; *Figure 4—figure supplement 1*). However, this repulsion is not restricted to these two resistance mutations, which are only 237 bp apart. Rather, when we visualized the distribution of signed LD between all bi-allelic missense SNPs around the ALS locus, and the two TSR mutations, we observed a predominance of positive association with the focal ALS Trp-574-Leu mutation (n = 19), but a negative association with the ALS Ser-653-Asn mutation (n = 34), across an extended 10 Mb region (*Figure 4A*). This long-range repulsion is not seen for other pairs of loci that are outliers for particularly strong repulsion within 500 bps (*Figure 4—figure supplement 2*). The divergence associated with these two TSR mutations segregating in Essex is also apparent with a genotype-based PCA—structure that is otherwise absent across the genome (*Figure 4B*, *Figure 4—figure supplement 3*). Together, this indicates that selection for these alternative resistance variants from ALS-inhibiting herbicides within Essex must occur through the competition of extended haplotypes up to ⅓ the size of this focal chromosome.

While the competition of haplotypes harbouring these TSR mutations may be important for shaping the distribution of resistance alleles across populations, the selective advantage of a given

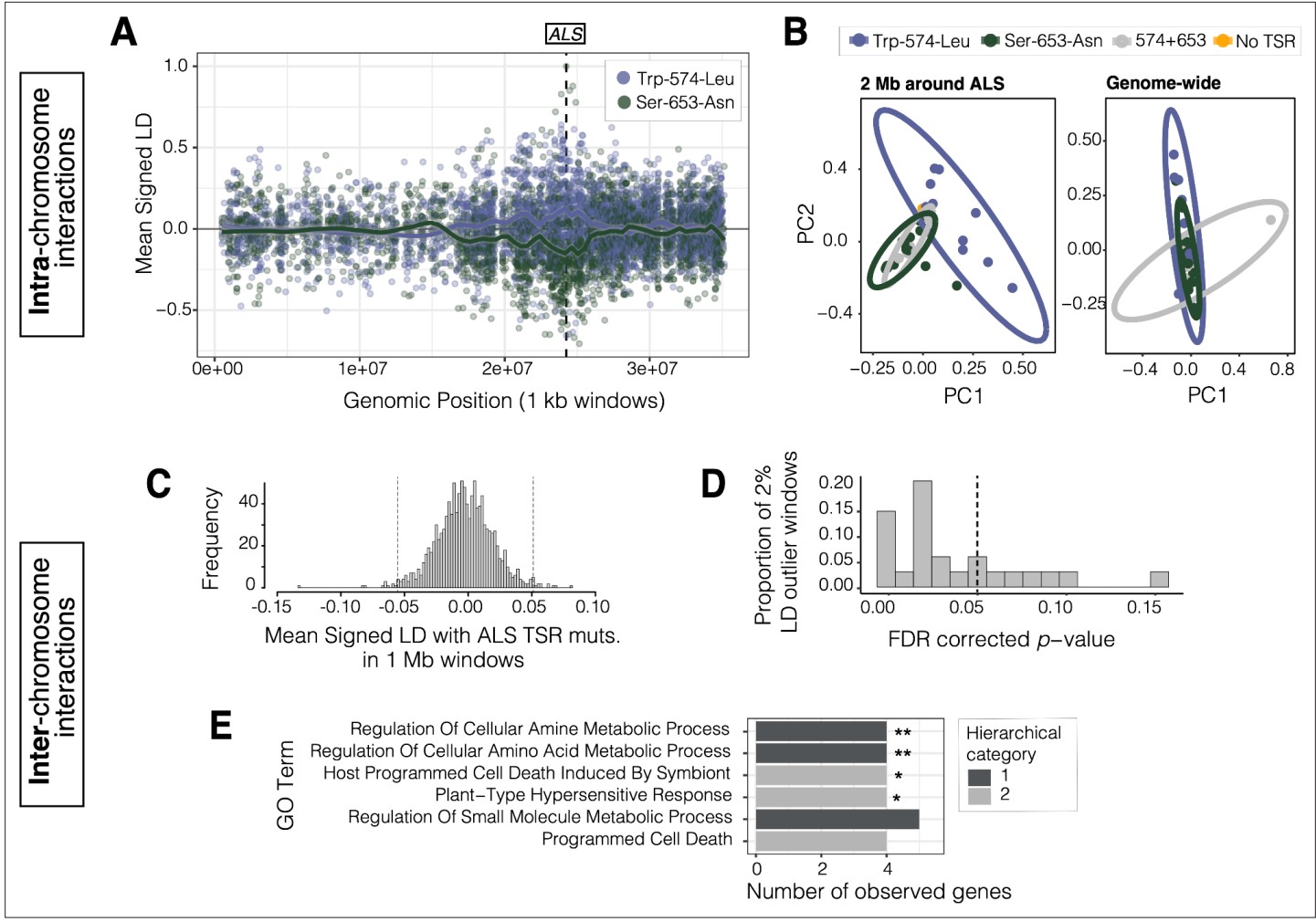

**Figure 4.** Signals of intra- and inter-chromosomal allelic interactions with target-site resistance mutations. (**A**) The breadth of haplotype competition between TSR mutations, illustrated by repulsion linkage disequilibrium (opposite signed LD, *r*) between two target-site-resistance mutations and bi-allelic missense SNPs surrounding them on Scaffold 11 in Essex. Each point shows mean LD in non-overlapping 10 kb windows. A smoothing spline shows that missense SNPs tend to be in positive LD with ALS Trp-574-Leu but negative LD with ALS Ser-653-Asn in Essex. (**B**) Signatures of population structure for 2 Mb around ALS compared to genome-wide, based on PCAs of genotypes in Essex. Ellipses represent 95% CIs assuming a multivariate distribution. (**C**) Distribution of mean signed LD of ALS TSR resistance mutations (ALS 574 or 653) with 1 Mb windows genome-wide in Essex, excluding the ALS containing Scaffold 11. Upper and lower 1% quantile indicated by dashed vertical lines. (**D**) Distribution of p-values from top 2% of genome-wide windows with high absolute signed LD with ALS TSR mutations, from permuting individual assignment within genomic windows and recalculating LD 1,000 times. (**E**) GO terms significantly enriched for biological process after FDR correction from the set of 348 genes mapping to the top 13, 1 Mb windows that show significantly extreme LD with ALS TSR mutations in Essex. Number of asterisks represent significance level after bonferroni correction (** = p < 0.01, * = p < 0.05).

The online version of this article includes the following source data and figure supplement(s) for figure 4:

**Source data 1.** Signed LD in 1 kb windows with two target-site-resistance mutations and bi-allelic missense SNPs surrounding them on Scaffold 11 in Essex.

**Source data 2.** Local PCA at 2 Mb around ALS, along with a genome-wide PCA (excluding the ALS containing scaffold 11), in Essex.

**Source data 3.** Signed LD of 1 Mb windows across the genome (aside from scaffold 11) with either ALS Trp-574-Leu and ALS Ser-653-Asn.

**Figure supplement 1.** Signed LD *(r)* between pairs of missense (nonsynonymous) and synonymous SNPs of similar frequency (minor allele frequency > 0.20) and physical distance ( < 500 bp) as ALS Trp-574-Leu and ALS Ser-653-Asn across the genome.

**Figure supplement 2.** Examples of the extent of extended repulsion elsewhere across the genome, for pairwise comparisons in *Figure 4—figure supplement 1* that were more extreme than observed.

**Figure supplement 3.** Signature of population structure for 10 Mb around ALS based on PCAs of genotypes in Essex.

**Figure supplement 4.** Correlation of signed LD *(r)* between two target-site resistance mutations, and synonymous or missense mutations across the genome.

TSR haplotype may also depend on other modifier loci across the genome. In particular, we might expect that individuals that have withstood many generations of herbicide applications due to large-effect resistance mutations may have also accumulated compensatory and tolerance-conferring mutations across the genome (stacking of resistance alleles; *Busi et al., 2013*; *ffrench-Constant et al., 2004*; *Kreiner et al., 2020*; *Petit et al., 2010*; *Preston, 2003*). Considering that haplotype competition seems to have manifested itself in patterns of signed LD within the ALS containing chromosome, we posited that physically unlinked modifiers of resistance (*Busi et al., 2013*; *ffrench-Constant et al., 2004*; *Kreiner et al., 2020*; *Petit et al., 2010*; *Preston, 2003*) could be identified due to strong linkage with focal TSR loci.

To identify such interactions in populations in Essex, we calculated mean signed LD between focal ALS Trp-574-Leu or ALS Ser-653-Asn mutations and bi-allelic missense alleles in 1 Mb non-overlapping windows across the genome in Essex (*Figure 4C*). We then focused on the upper 1% and lower 1% of windows with particularly extreme signed LD with either TSR mutation (24/1156 1 Mb windows). On each outlier window, we performed a permutation in which we randomized TSR allele assignment among individuals 1000 times, to test whether observed LD with resistance genotypes was more extreme than expected. Compared to the null expectation, the 1 Mb window with the strongest ALS TSR association showed a significant excess of positive inter-chromosomal signed LD with ALS Trp-574-Leu (one-tailed $p < 0.0001$, $r = + 0.068$) but negative signed LD with respect to the ALS Ser-653-Asn mutation ($r = –0.132$), consistent with repulsion between TSR alleles. Upon further inspection, this 1 Mb region is directly centred on a cytochrome P450 gene, CYP82D47, that has been implicated in conferring non-target site resistance in *Ipomoea purpurea* (*Leslie and Baucom, 2014*).

Of the 24 outlier windows, 13 had p-values consistent with significantly extreme LD with TSR loci after FDR correction at $\alpha = 0.05$ (*Figure 4D*). These 13 windows included 348 genes, 120 of which have *Arabidopsis thaliana* orthologs. These 120 orthologs were enriched for six GO biological processes belonging to two unique hierarchical categories after FDR correction, four of which were enriched even after Bonferroni correction: cellular amine and amino acid metabolic process, programmed cell death, and plant-type hypersensitive response (*Figure 4E*), seemingly directly related to the function of ALS—amino acid synthesis. Two particularly interesting examples from our set of genes in strong inter-chromosome LD with ALS resistance mutations are those encoding GCN2 (general control non-repressible 2) and KIN10 (SNF1 kinase homolog 10). Both proteins have been previously identified as playing key regulatory roles in response to herbicides, with GCN2 directly involved in homeostatic tolerance to ALS and glyphosate herbicides through regulating autophagy and amino acid signaling (*Faus et al., 2015*; *Zhao et al., 2018*). Similarly, KIN10, a key positive regulator of autophagy in *A. thaliana*, is activated in response to photosystem II herbicides (*Baena-González et al., 2007*; *Chen et al., 2017*; *Fujiki et al., 2001*).

## Discussion

The application of herbicides to manage agronomically important weeds has led to one of the best-studied examples of parallel evolution outside the laboratory, with target-site-resistance mutations to ALS-inhibiting herbicides identified in more than 150 species (*Heap, 2014*). We have studied the evolution of resistance mutations at two genes, ALS and PPO, from a genome-wide perspective across a large fraction of the range of one of the most problematic weeds in the US and Canada, *A. tuberculatus*. We found rampant evidence for both independent origins and gene flow, competition among resistant haplotypes, and the interaction of large-effect TSR mutations with physically unlinked alleles with resistance-related functions. These results paint a picture of the rise, spread, and fate of adaptive alleles in the face of extreme selection, with important implications for the management of herbicide-resistant agricultural weeds.

### Repeated origins and the spread of resistance via gene flow

We detected strong evidence for parallel evolution to herbicides within *A. tuberculatus* agricultural weed populations at multiple levels. Target-site mutations conferring resistance to PPO and ALS herbicides in the sampled population were found at seven distinct codons, with nine distinct variants, three of which are common. These three common mutations have arisen repeatedly seven times across our sampled populations, based on ARG inference (*Table 1*, *Figure 2*), consistent with the

largely soft selective sweep signals we observe at a regional scale (*Figure 1*). ARGs have seen limited implementation outside of human populations for examining patterns of local adaptation but recently have been used to infer the evolutionary processes that govern islands of differentiation across birds (*Hejase et al., 2020*). In comparison to gene trees that illustrate the average coalescent history of these genomic regions (*Figure 2—figure supplement 1*), we show that accounting for intragenic recombination through ARG inference has been extremely valuable for further resolving independent origins of adaptive mutations.

From a mutation-limited view of adaptation, the extent of parallelism in target-site herbicide resistance that we observe here is particularly extreme. However, given the estimate of North American *A. tuberculatus* $4N_e\mu \sim \Theta_\pi = 0.041$ for putatively neutral sites (*Kreiner et al., 2019*), a new TSR mutation at any of the eight adaptive ALS mutations should arise at a rate of $\Theta_\pi/2$—once every six generations (i.e. 0.041/2 x 8 known TSR loci = 0.164 TSR mutations per generation; see also *Charlesworth, 2009*; *Karasov et al., 2010*). Furthermore, the estimated $\Theta_\pi$ we use to infer the rate of new mutations may even be an underestimate given contemporary population size, which may be closer to census size than long-term estimates of $N_e$ from neutral polymorphism, should determine the mutational supply for rapid adaptation under models of evolutionary rescue (*Bell, 2013*; *Karasov et al., 2010*; *Kreiner et al., 2018*; *Neve et al., 2014*). Indeed, if we modify this value to reflect the contemporary estimate of $N_e$ over the last 50 years (~$8 \times 10^8$) and assume an *A. thaliana* mutation rate of $7 \times 10^{-9}$ (*Ossowski et al., 2010*; *Weng et al., 2019*), our $\Theta$ becomes >1 and a new mutation at any TSR codon should arise every generation—consistent with the remarkably parallel mutational origins we describe here. From these inferences, parallelism in simple target-site herbicide resistance adaptation in *A. tuberculatus* appears to be on par with prokaryotic adaptation and pesticide resistance adaptation in *Drosophila melanogaster*, where population sizes on the order of $\Theta \sim 1$ facilitate adaptation to occur rapidly, without being limited by mutational input at single sites (*Karasov et al., 2010*).

In the context of such extreme recurrent evolution, we still find an important role of gene flow in the spread of herbicide resistance across the range. Not only do agricultural regions and populations within them harbour multiple origins of TSR, but distinct recombinational units harbouring these mutational origins also map to many populations (*Figure 2B*). The widespread impact of gene flow is further consistent with our inference of near panmixia for all but two resistant lineages, although our permutation test is limited in power for rare origins. In part, widespread movement of *A. tuberculatus* and TSR variants across the North American range is likely to reflect the massive recent expansion we see here (*Figure 3A*)—population size increasing by four orders of magnitude over the last 100 years. This expansion also corresponds well with *A. tuberculatus*'s contemporary agricultural association (*Sauer, 1957*), suggesting that agronomic regimes are likely to have in large part facilitated the success of this weed species. Thus, both extreme mutational parallelism and a complex network of haplotype sharing, via gene flow and colonization, characterize the distribution of herbicide resistance across our sampled agricultural populations.

While the role of repeated origins and widespread gene flow we characterized here fit well with the cosmopolitan and highly convergent nature of herbicide resistance adaptation, the patterns we observe may be influenced by the sensitivity of ARG inference to both phasing quality and fine-scale recombination rate variation. We performed a two-step phasing method, performing population-level phasing with SHAPEIT4, a powerful up-to-date method (*Delaneau et al., 2019*), after performing read-backed phasing with WhatsHap (*Martin et al., 2016*). Nonetheless, phase switching remains a challenge for haplotype inference in naturally occurring populations in lieu of long-read population resequencing. Phase-switching between haplotypes is likely to be interpreted as a recombination event during ARG inference, however, by explicitly modeling how these 'recombinational' units relate to one another, ARG inference should still be better able to resolve independent origins of adaptive haplotypes than traditional reconstruction methods. Nonetheless, to adjust for the potential phase-switching that may artificially inflate recombination rates, we ran ARGweaver over three magnitudes of recombination rate values ($r = 10^{-7}$ to $10^{-9}$), as well as two resolutions of discrete time steps (t = 20, 30) (*Figure 2—figure supplement 2*), to find the parameter values that maximized the likelihood of our phased data ($r = 10^{-8}$ and t = 30). Even across less likely parameter values, we find consistent support for multiple origins across all TSR loci (*Figure 2—figure supplement 3*).

## The selective history of target-site resistance lineages

Detrimental effects of monogenic resistance mutations as a result of pleiotropic tradeoffs and fluctuating ecological selective pressures (*Lenormand et al., 2018*) have led to the question of whether such costs could be leveraged to prevent the persistence of resistance mutations (*Vila-Aiub, 2019*). We were able to evaluate evidence for TSR alleles that predated the onset of herbicide usage, and thus learn about how long these alleles persist, by rescaling allelic age estimates by the geometric mean effective population size estimate over the last 50 years (*Figure 3*). We found evidence for one mutational lineage vastly predating the onset of ALS herbicide use, arising nearly 350 years ago; however, this lineage only showed 45% support for a monophyletic origin across MCMC iterations of the ARG inference. By and large, our results suggest that the ALS resistance mutations we have sampled arose repeatedly, soon after the introduction of ALS-inhibiting herbicides in the 1980s. The haplotypes on which they arose have since been subject to widespread gene flow, which can facilitate a rapid response to selection across a large geographic range. In comparison to these nonsynonymous substitutions within the ALS gene, the single-codon deletion that confers resistance to PPO-inhibiting herbicides appears to be much younger, estimated to have only arisen within the last 3–4 years prior to population sampling.

While these allele age estimates provide a powerful test of the extent of adaptation from standing variation versus from new mutations, and the timescale over which resistance mutations may persist, these estimates are an approximation based on the geometric mean $N_e$ over the last 50 years, and do not fully account for the monumental population expansion this species shows. Furthermore, this rescaling depends on the accuracy of our $N_e$ estimate through time, as inferred by Relate (*Speidel et al., 2019*). Previously, we have also implemented δaδi (*Gutenkunst et al., 2009*), which uses the site frequency spectrum, to infer demographic history in these samples and similarly found evidence for large recent population expansion in a two epoch model (*Kreiner et al., 2019*). Relate is a better method for $N_e$ inference in this scenario, however, as it provides estimates of $N_e$ on contemporary timescales (0–100 years ago). Further, it has been shown to be accurate in the face of phasing error, with only a slight overestimation of $N_e$ on recent timescales (*Speidel et al., 2019*). Nonetheless, this slight bias towards larger $N_e$ estimates in the face of phasing error may suggest that our rescaled allele ages tend to underestimate the true age of these mutations.

Given the challenges of allele age estimation, we also used a conceptually related approach, testing for evidence of selection over two different timescales with a tree-based statistic that is robust to population size misspecification. Reassuringly, this tree-based test clearly shows evidence of selection since a mutation arose (consistent with de novo adaptation) for the history of four out of seven lineages. However, this test further demonstrates that these lineages are even more likely to have been under selection on more recent timescales, with rank order shifts in support across resistant lineages as compared to support for consistent selection over their history. Thus, while by our estimates, adaptation to PPO- and ALS-inhibiting herbicides relies predominantly on de novo mutation, spatially and temporally varying selection has resulted in muted signals of selection over the course of many mutational lineages' histories—selection that has intensified over timescales even more recent than the onset of herbicide use in some geographic regions.

While the intensification of herbicide use over the last half-century may be one source of temporally varying selection, rotating herbicide and cropping regimes may also contribute to fine-scale fluctuations in selection for or against particular TSR mutations. For example, in corn and soy production systems (where the focal crop alternates each season), PPOs were typically used only in soy, whereas ALS herbicides were heavily used in both crops (*Salas et al., 2016*; *Tranel and Wright, 2002*). Thus, the lower net-positive selection from PPO-inhibiting herbicides along with their recent resurgence in popularity (*Tranel, 2021*) may explain the more recent origins and thus shorter persistence of the PPO ΔGly210 alleles. Similarly, while the ALS Trp-574-Leu mutation tends to confer resistance broadly across all ALS herbicides, the ALS Ser-653-Asn mutation confers resistance to fewer types of ALS-inhibiting herbicides, which also happen to be used more commonly in soy (*Patzoldt and Tranel, 2017*). This may contribute to the relatively lower frequency of ALS Ser-653-Asn compared to ALS Trp-574-Leu across the *A. tuberculatus* range, or even suggest that a lack of both focal crop and herbicide rotation facilitated strong recent selection on the high-frequency Ontario ALS Ser-653-Asn lineage.

## Intra and inter-locus allelic interactions shape the history of TSR mutations

The outcome of parallel adaptation in a continuous species range has been thoroughly described by *Ralph and Coop, 2010*. When the geographic spread of an adaptive mutation is migration limited, partial sweeps for parallel adaptive mutational origins that occur in distinct geographic regions will be common. However, as 'waves of advance' of these distinct mutational origins expand, eventually coming into contact, beneficial haplotypes will compete on their way to fixation (*Ralph and Coop, 2010*). Given our evidence for highly parallel TSR adaptation across the range, we expect that this scenario fits the evolution of resistance particularly well. The widespread gene flow we observe sets up a scenario where independent origins of TSR mutations have now met and must interact. Under such a scenario, further background-dependent fitness effects, additive and epistatic interactions with both physically linked and unlinked loci, may shape the success of particular mutational lineages.

The coexistence of mutations of independent origin under strong selection in a given locale, notably of the ALS Trp-574-Leu and ALS Ser-653-Asn mutations in Essex, has allowed us to observe the signature of such intra-locus interactions. The 10 Mb stretch of repulsion and strong haplotypic divergence we observe (*Figure 4A and B*) likely reflects the unique spatial origins of each focal ALS TSR mutation and local selection. However, these mutations now co-occur across many agricultural regions. The ongoing repulsion of these alleles, combined with selection being constrained to act on independent allelic combinations (*Otto, 2021*), means that further adaptation to ALS-inhibiting herbicides must occur through the competition (sensu *Mather, 1969*) of these extended resistance haplotypes nearly 10 Mb long.

One question that arises based on our observations of haplotypic competition is why recombination has not eroded the signal of LD, and further, why we do not observe a single haplotype with both ALS mutations? The former is clearly not because of a lack of opportunity for recombination, since trans-heterozygous individuals that carry both mutations in opposite phases are not uncommon, and both mutations segregate at considerable frequencies. However, it is possible that recombinant double mutants are only rarely generated through recombination. The local LD-based population recombination rate estimate of $\rho$ = 4 $N_e r$ = 0.0575 in a region of 100 kb on either side of ALS implies 3.4 new recombination events per generation in the distance between these two mutants (237 bp x (0.057/4)). Accounting for the local frequencies of ALS Trp-574-Leu and ALS Ser-653-Asn alleles (0.29 × 0.44), in a panmictic population, ~ 1 of these recombination events should generate a double resistant mutant every other generation. However, recombination is preferred in promoter regions (*Kent et al., 2017*; *Sandler et al., 2020*), suggesting that these calculations may overestimate intragenic levels of recombination and thus the opportunity to generate recombinant resistant haplotypes.

Understanding the fitness consequences of these alleles, both separately and in combination, will determine how this competition will be resolved. If one allele confers significantly greater resistance (and assuming similar costs in the absence of herbicides), we may expect it to eventually reach fixation, outcompeting the alternate haplotype. Alternatively, if the alleles are selectively neutral with regard to each other, stochastic processes may predominate (*Ralph and Coop, 2010*). Given enough time, however, recombination should create haplotypes carrying both TSR mutations. It is possible that the joint effect of non-synonymous mutations at codon 574 and 653 on ALS protein structure results in negative epistasis, which would keep such double resistant haplotypes rare (as observed for antibiotic resistance alleles, *Porse et al., 2020*). Alternatively, additive or positive epistatic effects between these mutations would favour fixation of a double mutant haplotype, suggesting that the current observed haplotypic competition is in fact a form of Hill-Robertson interference (where the rate of adaptation is slowed due to linkage) which can eventually be resolved through recombination (*Cooper, 2007*; *Hill and Robertson, 1966*; *Otto, 2021*).

While we find that intra-chromosomal interactions such as competition between alleles have influenced the selective trajectory of individual TSR alleles, we were also interested in the extent to which interactions with physically unlinked loci may have facilitated herbicide resistance evolution. We find evidence that selection on Essex haplotypes containing ALS TSR mutations has likely been affected by such second-site modifiers (*Figure 4C–E*). We find particularly extreme signed LD between TSR mutations and alleles on different chromosomes. LD between resistance alleles on different chromosomes has been interpreted as epistasis (*Gupta et al., 2021*), but LD between alleles that are not physically

linked may also result from the effects of additive alleles with correlated responses to selection (e.g. the stacking of resistance alleles; *Busi et al., 2013*).

Alleles in windows on different chromosomes with the strongest evidence of linkage with ALS target-site resistance mutations function in biological processes related to known organismal responses to ALS herbicides. ALS-inhibiting herbicides disrupt biosynthesis of branched amino acids, and a rapid response after exposure leads to amino-acid remobilization through enhanced protein degradation (autophagy) and reduced synthesis (*Orcaray et al., 2011*; *Trenkamp et al., 2009*; *Zhao et al., 2018*; *Zulet et al., 2013*). We observe significant enrichment for functions in both amino acid production and cell death. These alleles may thus provide additional levels of tolerance on the large-effect TSR background, by compensating for homeostatic disturbances caused by ALS exposure or the potential costs of large-effect resistance mutations (as seen for antibiotic resistance, *MacLean et al., 2010*).

## Conclusion

In conclusion, adaptation to herbicides and the emergence of well-characterized target-site resistance mutations provide a powerful system for characterizing rapid and repeated evolution in plant populations, as well as the consequences of extreme selection on local and genome-wide patterns of diversity. Studies of herbicide resistance evolution have highlighted how extreme selection can modify life-history and plant mating systems (*Kuester et al., 2017*; *Van Etten et al., 2020*) and vice versa (*Kreiner et al., 2018*), as well as the role of small- versus large-effect mutations (or monogenic versus polygenic adaptation) (*Kreiner et al., 2021*; reviewed in *Délye, 2013*; *Powles and Yu, 2010*), costs of adaptation under fluctuating environments (*Vila-Aiub, 2019*; *Vila-Aiub et al., 2009*), and mutational repeatability (e.g. *Heap, 2014*; *Menchari et al., 2006*) (see *Baucom, 2019*). The work here contributes to this literature by characterizing not only the parallel origins and spatial distribution of target-site-resistance alleles across a broad collection of agricultural populations, but also heterogeneity in their evolutionary history and key contributing processes, such as fluctuating selection, haplotype competition, and cross chromosomal linkage with putative resistance alleles.

From a practical perspective, this work highlights *A. tuberculatus* as a worst-case scenario for controlling the evolution of herbicide resistance and containing its spread. Large census and effective population sizes facilitate extreme convergence for repeated selection of identical nucleotide polymorphisms conferring resistance. One key priority for thwarting new resistance mutations from arising and spreading should thus be containing *A. tuberculatus* population sizes as much as possible. Our findings suggest that most resistance mutations are of very recent origin and that they can persist for several decades with average herbicide usage. Across growing seasons, our best hope is to reduce the fitness advantage of resistant types by decreasing reliance on herbicides, rotating herbicide chemical compounds and mode-of-actions, and using herbicide mixtures. However, one must also be wary of neighboring weed populations. Gene flow of resistance mutations across the range is widespread, facilitating a rapid response to selection from herbicides, potentially even in naive populations. In particular, we find strong regional patterns in the distribution of resistant lineages, suggesting coordinated herbicide management regimes across farms, land-use planning, and hygienic machine-sharing will be important tools for efficient control of herbicide-resistant weeds.

## Materials and methods
### Amaranthus tuberculatus sequence data

Sequencing and resequencing data were from a published study (*Kreiner et al., 2019*). Whole-genome Illumina sequencing data are available at European Nucleotide Archive (ENA), while the reference genome and its annotation are available on CoGe (reference ID = 54057). The analyses in this paper focus on herbicide resistance in 158 agricultural samples, collected from eight fields with high *A. tuberculatus* densities across Missouri and Illinois in the Midwestern US (collected 2010), and from newly infested counties in Ontario, Canada, Walpole Island, and Essex County (collected 2016). The eight Midwestern US populations previously had been surveyed for resistance to glyphosate (*Chatham et al., 2017*). Ten additional samples collected from natural populations in Ontario, Canada are also included, but only for tree-based inference. These samples have been recently analyzed with respect to the convergent evolutionary origins of amplification of the glyphosate-targeted gene,

5-enolpyruvylshikimate-3-phosphate (*Kreiner et al., 2019*), as well as the polygenic architecture of glyphosate resistance (*Kreiner et al., 2020*).

## SNP calling and phasing genotypes

Filtered VCFs were obtained from *Kreiner et al., 2019* for all analyses. Briefly, freebayes-called SNPs were filtered based on missing data ( > 80% present), repeat content, allelic bias ( > 0.25 and < 0.75), read paired status, and mapping quality (> Q30). Six individuals were removed due to excess missing data, leaving 152 for agricultural and 10 natural samples for further analyses.

Known TSR mutations were assayed for presence/absence in our set of 162 *A. tuberculatus* individuals. At the time, that meant checking for known TSR mutations at eight ALS codons (*Yu and Powles, 2014*), three PPO codons (*Giacomini et al., 2017*; *Rousonelos et al., 2017*; *Varanasi et al., 2017*), one PsbA (conferring resistance to photosystem II inhibitors) codon (*Lu et al., 2019*), and three EPSPS codons (*Perotti et al., 2019*). To assay these mutations in our samples, we referred to the literature on previously verified TSR mutations, extracting the sequence surrounding a given focal TSR mutation, and BLAST (*Altschul et al., 1990*) searched our reference genome to locate its position.

To phase genotypes into haplotypes, we first used WhatsHap (*Martin et al., 2016*), which performs readback phasing, and passed these phase sets to SHAPEIT4 (*Delaneau et al., 2013*) which further phases haplotypes based on population-level information. Since phasing is very sensitive to data quality, we also applied a more stringent threshold of no more than 10% missing data for each SNP. SHAPEIT4 also requires a genetic map; with no recombination map for *A. tuberculatus* yet available, we used LDhat to infer recombination rates across the genome in our samples (as in *Kreiner et al., 2019*). Specifically, we used the interval function to estimate variable recombination rates within each of the 16 chromosomes of the pseudoassembly, using a precomputed lookup table for a θ of 0.01 for 192 chromosomes. We then converted rho estimates to genetic distance-based recombination rates ($100/4N_er$; $N_e$ = 500,000), and used a monotonic spline to extrapolate genetic distance to each SNP in our VCF. We provided SHAPEIT4 an effective population size estimate of 500,000, inferred from previous demographic modeling in dadi (*Kreiner et al., 2019*).

## Tree inference

Gene trees were inferred based on haplotypes within focal target-site genes (ALS and PPO), and 5 kb on either side around them. This resulted in 296 SNPs and 622 SNPs for inference of ALS and PPO trees respectively. Using the phased data around these genes, we first converted each phased haplotype to FASTA format, performed realignment with clustal omega (*Madeira et al., 2019*), and then ran clustal-w2 (*Rédei, 2008*) to infer the maximum likelihood tree, once for each gene, with 1,000 bootstraps. We then plotted mutational status for each focal TSR mutation (ALS Trp-574-Leu, ALS Ser-653-Asn, and PPO ΔGly210) at each tip of both gene trees using ggtree (*Yu, 2020*).

We ran ARGweaver (*Hubisz and Siepel, 2020*; *Rasmussen et al., 2014*) on a region of 20,000 SNPs centered between the ALS and PPO genes on Scaffold 11 across our phased haplotypes. We used the settings -N (effective population size) 500,000 -m (mutation rate) $1.8e^{-8}$ --ntimes (estimated timepoints) 30 --maxtime (max coalescent time) $100e^3$ -c (bp compression rate) 1 -n (MCMC samples) 1,250. We used an effective population size of 500,000, based on the best fitting demographic model previously inferred from this dataset with δaδi (*Gutenkunst et al., 2010*; *Kreiner et al., 2019*). To account for any bias introduced by phasing error in the form of inflated recombination rates, we ran multiple iterations of ARGweaver at varying constant recombination rates, from $r = 7e^{-7}$ to $7e^{-9}$, drawing our inferences from the parameter values that maximized the likelihood of observing our data ($r = 7e^{-8}$). Similarly, we also tested out two parameter values of --ntimes (20, 30), and used t = 30 for our inference, together with $r = 7e^{-8}$. We then extracted the most likely ARG sample from the MCMC chain (sample 1240), and the local trees corresponding to each of our three focal TSR mutations using arg-summarise. The arg-summarize function of ARGweaver was used to estimate the mean and 95% confidence intervals of the age of each mutational origin (based on clusters inferred from the most likely trees in the previous step) across all MCMC samples of the ARG. Since by default, arg-summarise --allele-age will infer the age of only the oldest allele under a scenario of multiple origins, we subsetted the dataset one mutational origin at a time (including all susceptible haplotypes) to obtain age estimates for each origin. ARGweaver runs on phased bi-allelic SNP data. Therefore to obtain a tree for the PPO ΔGly210 deletion, we recoded the VCF entry at this indel as an SNP,

changing the reference allele at this site to a single bp record matching the reference call at the start position (G) with an alternate single base-pair call (A).

As a test for panmixia, we performed a permutation test of the proportion of a mutational lineage (i.e. haplotypes belonging to a particular origin) mapping to Ontario (Essex and Walpole) compared to the Midwest. To do so, we randomized haplotype assignment to either Ontario or the Midwest, and recalculated the proportion of haplotypes belonging to each origin found in each region, 1000x. We then calculated the 95% CI of the proportion of haplotypes mapping to either Essex or the Midwest, as the null expectation under panmixia. When our observed value for a given origin exceeded the null expectation, we took this as significant evidence for stratification in the geographic distribution of a mutational lineage.

## Coalescent tree-based tests for selection

RELATE (*Speidel et al., 2021*; *Speidel et al., 2019*), a scalable method for estimating ARGs across large genomic datasets implements a tree-based test for detecting positive selection (*Griffiths and Tavaré, 1998*; *Speidel et al., 2019*). Under the standard coalescent model (i.e. assuming selective neutrality of mutations), the number of descendants in a particular lineage is exchangeable. Thus, one can compute the probability of some observed skew in the number of descendants using the hypergeometric distribution (*Griffiths and Tavaré, 1998*; *Speidel et al., 2019*). This approach gives us a p-value for this skew under the null (i.e. no selection). We did this on an origin-by-origin basis, comparing the rate of allele frequency change across the tree for a focal resistant lineage compared to all other susceptible haplotypes across the tree. Since this statistic is simply based on the order of coalescents, rather than branch lengths, it should be robust to misspecified effective population sizes used to infer our ARG (*Speidel et al., 2019*). Since RELATE assumes an infinite sites model and thus is unsuitable for testing hypotheses about multiple origins, we performed our own implementation of this method for trees outputted from Argweaver (*Rasmussen et al., 2014*).

Briefly, the statistic works as follows. Let $f_N$ be the number of carriers of our focal mutation in the current day, $N$ be the total present-day sample size, and $k_S$ be the number of susceptible lineages present when the mutation increases in count from 1 to 2. We sum each individual probability that a mutation spreads to at least a given frequency, from $f_N$ to $N$-$k_s$ + 2.

$$p_{R,de\,novo} = \sum_{f=f_N}^{N-k_S+2} \frac{(f-1)\binom{N-f-1}{k_S-3}}{\binom{N-1}{k_S-1}} \tag{1}$$

The null hypothesis, that allele frequency change occurred under drift, is rejected when this one-side p-value is sufficiently small (i.e. $\alpha < 0.05$), implying selection has governed the spread of this mutation since it first arose.

We also modified this statistic to test for selection on more recent timescales, and thus the scenario of adaptation from standing genetic variation. Here, we need to define $k_R$, the number of resistant lineages at some time ($t$) before the present day, in addition to $k_S(t)$.

$$p_{R,sgv} = \sum_{f=f_N}^{N-k_S+2} \frac{\frac{f-1}{(k_R(t)-1)}\frac{N-f-1}{(k_S(t)-1)}}{\binom{N-1}{k_R(t)+k_S(t)-1}} \tag{2}$$

The null hypothesis that the frequency change (between the current day and some time in the past more recent than when the mutation first arose ($t$) happened under random drift (and hence no selective pressures)) is rejected if this p-value is sufficiently small.

## N$_e$ estimation through time

We used RELATE 1.1.6 to estimate tree sequence from distinct recombinational units across the genome from our phased dataset. Relate requires polarized ancestral allele calls, such that alternate alleles represent the derived state. To do so, we performed a multiple alignment of our *A. tuberculatus* genome to *A. palmeri* (*Montgomery et al., 2020*) using lastz (*Harris, 2007*), retained the best orthologous chain from the alignment, and extracted variant sites. We modified the *A. tuberculatus* reference genome with the derived allele states from our multiple alignment, using this modified reference to polarize allele calls. On each chromosome, we then ran RelateParallel.sh ––mode All, using the output from all chromosomes to first estimate mutation rate (RelateMutationRate ––mode

Avg), reestimate branch lengths with this updated mutation rate (ReEstimateBranchLengths), and then lastly estimate population size through time (EstimatePopulationSize.sh). Population sizes were estimated from 0 years ago to 10,000,000 years ago, in epochs timesteps of $10^{0.25}$ years, to obtain particularly fine-scale estimates in the recent past.

## Selection scans and LD-based analyses

The phased data used as input for ARGweaver was also used to extract selective sweep summary statistics in selscan (*Szpiech and Hernandez, 2014*). In selscan, we estimated both XPEHH (*Sabeti et al., 2007*), in this case, the difference in the integrated extended haplotype homozygosity between resistant and susceptible haplotypes, and mean pairwise difference estimates. For both of these statistics, we provided LD-based recombination maps, inferred from LDhat. Because some individuals in Essex always carried at least one resistant ALS haplotype through either mutations at ALS Trp-574-Leu or ALS Ser-653-Asn, to compare patterns of selection associated with resistance and susceptibility, these statistics were calculated at the haplotype, rather than individual level. For each independent origin as inferred from ARGweaver, we similarly inferred XPEHH, as well as H12 (*Garud and Petrov, 2016*; *Garud and Rosenberg, 2015*) across the chromosome containing ALS.

We used plink (v1.90b3.46) (*Purcell et al., 2007*) to calculate both signed LD (*r*) between each focal TSR mutation and missense mutations on the same chromosome, and with all other bi-allelic missense SNPs across the genome. We used nonsynonymous SNPs as we expected them to be less influenced by population structure and admixture (*Good, 2020*) compared to synonymous SNPs, but present the correlation between genome-wide LD with synonymous and nonsynonymous SNPs in *Figure 4—figure supplement 4*. We performed these calculations with respect to a given TSR mutation by using the `--ld-snp` options to specify a focal mutation. To visualize patterns of signed LD between TSR mutations and other missense SNPs, we split the genome into non-overlapping 10 kb windows and calculated the average LD among all SNPs in each window. All LD calculations were polarized by rarity (e.g. minor alleles segregating on the same haplotypes were regarded as being in positive LD). In Essex, despite being considerably common, both ALS 574 and ALS 653 had a frequency less than 50%, so LD values with all missense alleles for each of these focal TSR mutations are directly comparable. As another visualization of the haplotype structure in this region, we conducted two genotype-based PCAs using the R package SNPrelate (*Zheng et al., 2012*), for genotypes spanning 2 Mb around ALS, 10 Mb around ALS, and on genome-wide genotypes.

To test whether the negative LD we observed between ALS Trp-574-Leu and ALS Ser-653-Asn was particularly extreme in Essex, we compared this value to pairs of either nonsynonymous or synonymous SNPs of similar frequency (minor allele frequency >0.20) and physical distance apart (< 500 bp). To test whether the top 2% of 1 Mb windows of nonsynonymous SNPs with particularly low or high signed LD with ALS TSR mutations (either ALS Ser-653-Asn or ALS Trp-574-Leu) was significantly different from the null expectation, we used a permutation approach whereby we randomly shuffled the assignment of the focal ALS TSR mutation between all individuals and calculated mean LD (with respect to the permuted TSR mutations) in the window of interest. We repeated this permutation 1000 times to generate a null distribution for comparison to the real average signed LD value of each region. This permutation test explicitly evaluates whether a TSR mutation and missense mutations in a focal window are more likely to be found together in the specific set of individuals containing the focal TSR mutation than any other set of individuals of the same size. Thus, this test is robust to variance in data quality across windows. The proportion of permuted observations with a mean absolute signed LD exceeding the observed signed LD was taken as the two-tailed p-value for cross-chromosome LD. Lastly, we found the intersect of these windows with the closest gene according to our genome annotation, and found their *A. thaliana* orthologs (*Emms and Kelly, 2015*). We used the set of *A. thaliana* orthologs found across all 13 significantly enriched 1 Mb windows in a Gene Ontology (GO) Enrichment analysis for biological processes.

## Acknowledgements

We gratefully acknowledge discussion and feedback on the manuscript from Matt Osmond, Aneil Agrawal, Rob Ness, Tyler Kent, and Sally Otto, and our two reviewers, Pleuni Pennings and Philipp Messer. Thanks to Leo Speidel and Melissa Hubisz for input on RELATE and ARGweaver implementation, and Peter Sikkema and lab for their support on sampling Ontario populations. JMK was supported

by the Rosemary Grant Advanced Award from the Society for the Study of Evolutionary Biology and NSERC PGS-D, DW was supported by the Max Planck Society, JRS and SIW were supported by Discovery Grants from NSERC Canada, and SIW was additionally supported by a Canada Research Chair in Population Genomics.

## Additional information

### Competing interests

Detlef Weigel: Deputy editor, eLife. The other authors declare that no competing interests exist.

### Funding

| Funder | Grant reference number | Author |
|---|---|---|
| Natural Sciences and Engineering Research Council of Canada | PGS-D | Julia M Kreiner |
| Society for the Study of Evolution | Rosemary Grant Advanced Award | Julia M Kreiner |
| Natural Sciences and Engineering Research Council of Canada | Discovery Grant | John Stinchcombe Stephen I Wright |
| Canada Research Chairs | Population Genomics | Stephen I Wright |
| Max Planck Society | | Detlef Weigel |

The funders had no role in study design, data collection and interpretation, or the decision to submit the work for publication.

### Author contributions

Julia M Kreiner, Conceptualization, Data curation, Formal analysis, Funding acquisition, Investigation, Methodology, Project administration, Software, Visualization, Writing – original draft, Writing – review and editing; George Sandler, Formal analysis, Investigation, Methodology, Visualization, Writing – original draft, Writing – review and editing; Aaron J Stern, Methodology, Software; Patrick J Tranel, Detlef Weigel, Data curation, Funding acquisition, Writing – review and editing; John R Stinchcombe, Data curation, Funding acquisition, Methodology, Supervision, Writing – original draft, Writing – review and editing; Stephen I Wright, Conceptualization, Data curation, Funding acquisition, Methodology, Supervision, Writing – original draft, Writing – review and editing

### Author ORCIDs

Julia M Kreiner (iD) http://orcid.org/0000-0002-8593-1394
George Sandler (iD) http://orcid.org/0000-0001-9420-1521
Aaron J Stern (iD) http://orcid.org/0000-0001-7368-5520
Detlef Weigel (iD) http://orcid.org/0000-0002-2114-7963
John R Stinchcombe (iD) http://orcid.org/0000-0003-3349-2964
Stephen I Wright (iD) http://orcid.org/0000-0001-9973-9697

### Decision letter and Author response

Decision letter https://doi.org/10.7554/eLife.70242.sa1
Author response https://doi.org/10.7554/eLife.70242.sa2

## Additional files

### Supplementary files

• Transparent reporting form

## Data availability

Sequencing data used in this paper were previously deposited in ENA under project number PRJEB31711, and reference genome is available on CoGe (reference ID = 54057). Code used to generate results in this paper is available at https://github.com/jkreinz/TSRevolution (copy archived at swh:1:rev:93ccf9e9d8d1fa25d1e4ef37a62bbb3c91df35dd).

The following previously published datasets were used:

| Author(s) | Year | Dataset title | Dataset URL | Database and Identifier |
|---|---|---|---|---|
| Kreiner JM, Giacomini D, Bemm F, Waithaka B, Regalado J, Lanz C, Hildebrandt J, Sikkema PH, Tranel PJ, Weigel D, Stinchcombe JR, Wright SI | 2019 | Multiple modes of convergent adaptation in the spread of glyphosate-resistant Amaranthus tuberculatus | https://www.ebi.ac.uk/ena/browser/view/PRJEB31711 | EBI, PRJEB31711 |

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
