## [Editor Report]

This paper studies the evolution of herbicide resistance in *Amaranthus tuberculatus*, a widespread agricultural weed. By illuminating how adaptive mutations arose and spread in this remarkable example of rapid human-induced adaptation, the study will be of interest to a broad audience, ranging from plant biologists interested in herbicide resistance to evolutionary biologists and population geneticists studying the fundamental factors and processes that govern rapid adaptation. The paper applies innovative population genetic methodology to support its primary finding that resistance mutations have evolved multiple times in parallel.

---

## [Decision Letter]

**Decision letter after peer review:**

Thank you for submitting your article "Repeated origins, gene flow, and allelic interactions of herbicide resistance mutations in a widespread agricultural weed" for consideration by *eLife*. Your article has been reviewed by 2 peer reviewers, including Philipp W Messer as the Reviewing Editor and Reviewer #1, and the evaluation has been overseen by Molly Przeworski as the Senior Editor. The following individual involved in review of your submission has agreed to reveal their identity: Pleuni S Pennings (Reviewer #2).

Essential revisions:

Both reviewers agree that this paper is interesting and should warrant publication in *eLife*, assuming that the authors can perform the following essential revisions. These points are laid out in more detail in the "recommendations for the authors" sections below.

1) Please add a discussion of what the results mean for preventing resistance in weeds, as pointed out by reviewer 2.

2) Reviewer 1 had some concerns about the robustness of ARG inferences to potential phasing errors. This should be discussed in more detail. Similarly, it should be tested whether results are robust to potential misspecification of the recombination rate at the resistance loci.

3) Please follow the suggestions provided by reviewer 2 regarding the analyses shown in Figure 2. It might be helpful to show the actual haplotypes.

4) Given that the finding of the recent population expansion is critical for the interpretation of allele ages, and therefore the question of whether adaptation occurred from SGV or de novo mutations, it would be reassuring if this result could be confirmed with a different inference method such as DaDi, stairway plot, or SMC++. If not possible, or if results are inconsistent, the claims about allele ages and sweep origins would need to be qualified accordingly.

5) Both reviewers were not entirely convinced by the finding of possible haplotype competition. Please clarify more precisely what you mean by interference / competition / etc. It should also be made more clear what the null expectation here is; for example, how often would ALS574 and ALS 653 be expected to occur together on the same haplotype in a model of free recombination?

*Reviewer #1 (Recommendations for the authors):*

The need for phased data for all RELATE, ARGweaver, and selection scan analyses raises some concerns about how potential phasing errors could affect the results. I feel the authors should at least discuss this. Ideally, additional analyses would be performed to test for such effects explicitly, although it is not clear to this reviewer what the most appropriate model would be. Maybe respective analyses were already performed in the papers that introduced the methods, in which case those results could simply be discussed here.

The recombination rate inference with LDhat could also be problematic at the resistance loci due to the presumably strong selection these loci have experienced, which may have affected LD patterns significantly. If, for example, sweeps have led to substantially elevated levels of linkage disequilibrium at these loci, LDhat would presumably interpret this as a lower recombination rate. My suggestion would be to run potentially affected analyses at the sweep loci not only with the recombination map inferred by LDhat, but also scenarios with a constant recombination rate set to a range of values, in order to test how robust the results would be to potential recombination rate misspecification.

The results from the demographic inference are intriguing and also play an important role for the interpretation of the results, as they provide the basis for the rescaling used to estimate both the population-level adaptive mutation rate and allele ages. If there would be a substantial error in the estimate of the recent effective population size, interpretations about adaptation from de novo mutation versus standing genetic variation could change quite dramatically. The demographic inference is currently based on the RELATE method. I wonder whether the authors have considered confirming these results with alternative methods to check how robust they are between different methods. For example, it would be interesting to compare this with SFS-based methods such as stairway plot and DaDi, or hybrid approaches such as SMC++. If the results from different methods agree, this would greatly increase trust in their accuracy. By contrast, if there are large discrepancies, possible reasons for this and potential implications for the interpretation of results would need to be discussed.

The arguably most speculative part of this study are the results on negative LD between common resistance mutations, which is interpreted by the authors as being caused by either haplotype competition, negative epistasis, or selective interference. Again, phasing could be somewhat problematic here. Maybe there are enough homozygotes in the data set that the authors could at least confirm that some of the findings hold even for unphased data? Also, I'm concerned that unknown population structure could potentially play into these results. This would be difficult to test, obviously, given that it's unclear what models one should test specifically. However, one question that I think could be more easily answered is how likely it is to find such negative LD at other genomic loci. Are the resistance loci truly genomic outliers in this regard? I hope the authors can add some discussion about whether they think population structure may or may not provide a potential alternative explanation for the observed negative LD.

*Reviewer #2 (Recommendations for the authors):*

I think this is a super interesting paper. It shows evidence for multiple origins (9) of drug resistance and also widespread migration (transmission) of resistance alleles between local populations. That in and of itself is worth publishing for me.

I feel like what is missing from the paper is a discussion of what these results mean for preventing resistance in weeds. I think that it means that when we want to prevent resistance, we need to be concerned with preventing mutation (control pop size / selection locally) and we need to be concerned with gene flow between populations (do not share equipment and staff?). I think the paper could benefit from more translation of the results to a non-evolutionary genetics audience. I also would love to see more easy-to-digest information about the relevant herbicides and the weed itself. This way, you can make the paper more interesting for evolutionary geneticists who have never thought about herbicide resistance and you can introduce those who are interested in plants and agriculture to evolutionary genetics.

In my opinion, some major improvements could be made to the presentation. Part of that is eye for detail in figures (e.g., use same colors and same notation throughout) and text (e.g., line 118 why mention psbA here, but nowhere else? Line 126 why mention glyphosate here? These things make the text hard to follow.…).

Some of the analysis on allele ages / evidence of recent selection should be presented differently before I am convinced.

The same is true for the analysis on clonal interference. This should start with clarifying what is meant by interference / competition / etc.

Instead of TSR I would write "resistance allele" – I think that will help other readers.

Figure 1: could Figure 1 be remade using the origins from Figure 2? Once you know that there are multiple origins, doing an analysis that ignores these origins doesn't make too much sense, I think. Can you use the same nomenclature as Table 1 and Figure 2?

Figure 2:

1. First, use ARG to find origins. Conclusion: there are multiple origins!

2. Then plot origins and fractions on a map. Conclusion: there is migration as well, though not panmixia (would it make sense to test that?).

3. Finally test for recent selection. Here I would love to see some kind of strength of selection statistic rather than just p-value.

For one allele there is evidence for recent selection, but not selection since origin (right?) – ALS 574#4. Could you show a zoom in for the tree with the 0.02% cut-off for that allele?

Conclusion of figure 2: multiple origins and gene flow are both important. For some alleles evidence that there has been selection since origin – suggests de novo evolution of resistance. But for some possibly standing genetic variation? I am not so convinced of that part of the analysis. Maybe it'd help to show haplotypes?

Suggestion: can you show haplotypes (like Harpak et al. on rats or Garud et al. on *Drosophila* or Williams and Pennings on HIV?)

Please label MidWest and Ontario. Could the pies have the same scale in the two regions?

Suggestion: Use the same color and nomenclature scheme throughout. For example, the 653_7 origin should have that name and color throughout the figure and other figures.

Figure 3:

The increase in pop size is clear and not surprising.

The allele ages are interesting. Could they be plotted?

I am not convinced that there is evidence for SGV for 653#7, because its age is only about 30 years according to the figure. 3B and 2C are hard to reconcile in my head.

Figure 4:

This is about interactions between ALS574 and ALS 653. These are 300-ish bp apart.

The resistance mutations never occur on the same haplotype which is surprising (how surprising? Could we get a prediction for how common this should have been if there was free recombination given the age of the alleles?).

Now, I'd say the next question is: how often do they occur in the same individual, given their commonality in each population. Like a HW-test. Is there selection against carrying two resistance alleles?

I am not entirely convinced that there is evidence for competition. Or maybe it is not clear to me what the authors mean exactly by competition / interference.

I am not sure my brain can follow Figure 4A – what would be the expectation here given multiple origins at both loci?

---

## [Author Response]

Essential revisions:Both reviewers agree that this paper is interesting and should warrant publication in eLife, assuming that the authors can perform the following essential revisions. These points are laid out in more detail in the "recommendations for the authors" sections below.1) Please add a discussion of what the results mean for preventing resistance in weeds, as pointed out by reviewer 2.

We have added a paragraph to the conclusions discussing the implications of this work for preventing the evolution and spread of resistance in weeds. In the introduction, we have expanded the background on the weed species and relevant herbicide use.

2) Reviewer 1 had some concerns about the robustness of ARG inferences to potential phasing errors. This should be discussed in more detail. Similarly, it should be tested whether results are robust to potential misspecification of the recombination rate at the resistance loci.

We performed extensive reanalysis to address this important issue, which greatly contributed to the time this work was under revision. We have rephased our entire dataset, using the most up-to-date phasing method, as well as performed ARG inference under multiple constant recombination rates (and time-step parameters) to focus our inference on the parameter set that maximizes the likelihood of observing our data and assess the extent to which the conclusions are robust to uncertainty in recombination rates. The main findings of this manuscript have remained consistent—we find evidence for multiple origins along with a widespread role of gene flow in their spread, and this is robust under multiple recombination rate values. We have also expanded our discussion of the limitations of these inferences (Lines 558-574, 594-607).

3) Please follow the suggestions provided by reviewer 2 regarding the analyses shown in Figure 2. It might be helpful to show the actual haplotypes.

We have included a visualization of the haplotypes corresponding to each origin as inferred from the ARG (Figure —figure supplement 4). As suggested, we also used the H12 (and the XPEHH statistic) to evaluate how these origins correspond to signatures of selection as you move across the genome (Figure 2—figure supplement 5,6), although given the nature of the algorithm used to infer these origins, we expect the signals to have been somewhat decayed due to both subsequent mutation and recombination.

4) Given that the finding of the recent population expansion is critical for the interpretation of allele ages, and therefore the question of whether adaptation occurred from SGV or de novo mutations, it would be reassuring if this result could be confirmed with a different inference method such as DaDi, stairway plot, or SMC++. If not possible, or if results are inconsistent, the claims about allele ages and sweep origins would need to be qualified accordingly.

In a previous publication, we used DaDi to perform demographic modelling of Amaranthus tuberculatus using the same WGS dataset. Qualitatively, this inference was consistent in that it inferred a large population expansion. However, a limitation of this demographic inference and most such inference methods in the context of this work, is that it is difficult to gain estimates over contemporary timescales, <100 or even 1,000 years ago. This is one reason why we were particularly excited about using Relate for such purposes. The expansion we observed over the last hundred years with Relate is consistent with the contemporary success of this plant as an agricultural weed, especially in light of reported densities up to 360 plants/m^2^ in heavily infested fields (Costea, Weaver, and Tardif 2005), and high levels of diversity in the species (Kreiner et al. 2019). The robustness of Relate has been tested under phasing error, for which it showed little bias in response to phase switching (Speidel et al. 2019). Nonetheless, we have qualified the interpretation of these results accordingly, especially as they pertain to allele age estimates (second paragraph of “The selective history of target-site resistance lineages” section of the Discussion; lines 594-607).

More generally, we recognize the difficulty in allele age estimation, and this was a major motivation for a complementary set of analyses we performed—a test for evidence of selection over two different timescales across the tree that is robust to effective population size misspecification (Figure 3B, Methods section “Coalescent tree-based tests for selection”). We used this tree-based test for selection to evaluate consistent selection over the entire history of a resistant lineage (conceptually similar to selection on a de novo mutation) as well as selection on timescales more recent than a mutation's origin. Consistent with our allele age estimates, these tree-based tests provide support for most alleles showing evidence of consistent selection since those alleles arose (de novo). In contrast to allele-age estimates, this test also provides evidence for fluctuating selection. While the majority of alleles show evidence for selection since they arose, we also observe generally stronger evidence of selection on even more recent timescales along with rank order shifts in significance across mutational lineages, suggesting the strengthening of selection from herbicides on recent timescales.

5) Both reviewers were not entirely convinced by the finding of possible haplotype competition. Please clarify more precisely what you mean by interference / competition / etc. It should also be made more clear what the null expectation here is; for example, how often would ALS574 and ALS 653 be expected to occur together on the same haplotype in a model of free recombination?

We have revised the writing of both the results and discussion of this topic greatly, to clarify what we mean by the intralocus interactions we describe as haplotype competition, and how it pertains to a model of the geographic spread of spatial parallel origins as in Ralph and Coop 2010 (Results, lines 337-339, 347-350; Discussion, lines 600-613). We suggest that the combination of parallel origins and subsequent geographic spread with strong selection has resulted in the competition of remarkable large resistant haplotypes (i.e., “genotypic selection” (Neher and Shraiman 2009), as opposed to fine-scale “allelic selection”) in the face of selection from herbicides.

We highlight that (in accordance with reviewer comments) distinct, spatial origins and strong local selection alone likely lead to the origin of the signature of negative LD and extreme structure that we observed between these local genotypes (Figure 4A, B). However, these mutations are no longer distinct in the space which they occupy, and coexist both globally and locally (Figure 2B). Thus, while the distinct spatial origins of these resistance mutations may have created a signal of haplotypic divergence, the fact that such a signal is only persistent in a large region around the ALS locus (locally in Essex where both mutations are at intermediate frequencies; Figure 4B), suggests that haplotypic competition between these mutations is present, and preserving this signal of divergence. In particular, we observe heightened LD across genotypes on this chromosome for a 10 Mb region, between genotypes associated with one resistance allele or another (i.e. repulsion). Further support for the independence of and thus competition between these coexisting alleles comes from haplotype-level inference. Despite observing 16 individuals that harbour both ALS Trp-574-Leu and ALS Ser-653-Asn globally, we observe no recombinant, double resistant haplotype. This lack of recombinant haplotypes is a strong violation of expectations under linkage equilibrium (^2^_df=1_ = 16.18, p = 5.77e^-5^). Given that these resistance mutations are selectively redundant to a number of ALS-inhibiting herbicides, the local patterns of genotypic and haplotypic structure we describe appears consistent with the gamut of the Ralph and Coop (2010) model: parallel spatially-distinct origins, strong selection, spread via gene flow, and haplotype competition.

Lastly, we speculate in the discussion about what additional processes could influence the ultimate outcome of this haplotypic competition based on potential fitness effects when these two alleles are combined onto the same background (Discussion lines 637-650).

Reviewer #1 (Recommendations for the authors):The need for phased data for all RELATE, ARGweaver, and selection scan analyses raises some concerns about how potential phasing errors could affect the results. I feel the authors should at least discuss this. Ideally, additional analyses would be performed to test for such effects explicitly, although it is not clear to this reviewer what the most appropriate model would be. Maybe respective analyses were already performed in the papers that introduced the methods, in which case those results could simply be discussed here.The recombination rate inference with LDhat could also be problematic at the resistance loci due to the presumably strong selection these loci have experienced, which may have affected LD patterns significantly. If, for example, sweeps have led to substantially elevated levels of linkage disequilibrium at these loci, LDhat would presumably interpret this as a lower recombination rate. My suggestion would be to run potentially affected analyses at the sweep loci not only with the recombination map inferred by LDhat, but also scenarios with a constant recombination rate set to a range of values, in order to test how robust the results would be to potential recombination rate misspecification.

We agree with these concerns and have performed extensive revisions to the methods and discussion (lines 558-574; 594-607) to address this comment.

The reviewer points out that issues with phasing, such as phase switching, may bias inferences from sweep scans and ARG inference. In response to this, we revisited our phasing method, and updated our approach from phasing with SHAPEIT2 to read-backed phasing in WhatsHap + population-level phasing with SHAPTEIT4. Intuitively, we find these updated methods have improved our inference as we are able to resolve a higher magnitude of difference in haplotype homozygosity and difference in mean pairwise diversity between susceptible and resistant haplotypes (e.g. XPEHH y-axis increased from 2 to 3). This improvement highlights that phase switching (e.g., swapping a resistant mutation onto a susceptible haplotype) can degrade signals of selection. However, it is clear that we are able to resolve strong signals of selection nonetheless, for example, in Essex where a selective sweep signal extends for ~10 Mb. We also performed a scan based on LD across genotypes, rather than haplotypes, which picks up on very similar patterns as our haplotype-based scans despite potential phasing bias (Figure 1).

While recombination rate inference is required for nearly all phasing software, we made sure to use a constant recombination rate for ARGweaver inference at our focal resistance loci. The ARGweaver manuscript (Rasmussen et al. 2014) specifically mentions how phasing errors will lead the algorithm to infer higher recombination rate values, although we see phase switching as less of an issue for the reconstruction of evolutionary history. In this case, high rates of phase switching will result in shortening of a given recombinational unit and thus less information supporting a focal tree. While this may be the case, we still find that this breadth of information is better able to resolve origins of resistance than gene-level inference.

Following your suggestions, we additionally tested ARGweaver over 3 orders of magnitudes of constant recombination rate values (as well as two preciseness of timesteps across the tree) to find the parameter sets that maximized the likelihood of observing our data *(*Results lines 211-233; Figure 2 – supplemental figure 2-4*)*. In particular, we focus our inference on t=30, r=10^-8^, which maximizes the likelihood of observing our data. However, we additionally show for other such parameter sets that a scenario of multiple origins is supported for all TSR loci (Figure 2 – supplemental figure 3), suggesting these inferences are robust to possible phase-switching inflating recombination rate. With a recombination rate either one order of magnitude lower or higher than maximizes our data, in all runs the origins inferred are completely consistent for ALS Ser-653-Asn and PPO210. Only for ALS Trp-574-Leu do we see some differences in the number of origins inferred, where using the most likely parameter set is able to resolve fewer origins (Figure 2 – supplemental figure 3).

Generally, while phasing error is a limitation of these ARG based analyses (discussed on lines 558-574), we find the application of ARGs to this dataset a valuable and powerful example of their use in genomic datasets. Compared to our gene-tree-based inference (Figure 2 supplemental figure 1), ARGs have been effective in bringing together origins that were separated out in the gene tree, whether they were being separated out as a result of recombination events subsequent to their origins or phase switching.

The results from the demographic inference are intriguing and also play an important role for the interpretation of the results, as they provide the basis for the rescaling used to estimate both the population-level adaptive mutation rate and allele ages. If there would be a substantial error in the estimate of the recent effective population size, interpretations about adaptation from de novo mutation versus standing genetic variation could change quite dramatically. The demographic inference is currently based on the RELATE method. I wonder whether the authors have considered confirming these results with alternative methods to check how robust they are between different methods. For example, it would be interesting to compare this with SFS-based methods such as stairwayplot and DaDi, or hybrid approaches such as SMC++. If the results from different methods agree, this would greatly increase trust in their accuracy. By contrast, if there are large discrepancies, possible reasons for this and potential implications for the interpretation of results would need to be discussed.

In a previous manuscript, we used DaDi to perform demographic inference in these samples of A. tuberculatus. Qualitatively, this two-epoch model showed a similar pattern, in that both major lineages of A. tuberculatus have experienced a recent population size expansion (Kreiner et al. 2019). However, a major limitation of this DaDi analysis, and other mentioned Ne size estimation software, is the timescale of this Ne reconstruction. In particular, we were drawn to Relate as it allows for Ne estimation over timescales relevant to herbicide use (the past ~50 years). To our knowledge, this remains a limitation of most other methods used for demographic inference. We have expanded our discussion of this Ne estimation and how bias in Ne estimates through time may influence allelic age estimates. In particular, while Relate has been shown to be relatively insensitive to phasing error (Speidel et al. 2019; Sup Figure 4), phase-switching may result in slightly overestimated effective population size, thus shifting our allele-age estimates to more recent timescales. Because of this limitation (discussed at lines 594-611), we also performed a conceptually related test robust to effective-population size misspecification, which examines evidence of selection across the tree over two different timescales.

The arguably most speculative part of this study are the results on negative LD between common resistance mutations, which is interpreted by the authors as being caused by either haplotype competition, negative epistasis, or selective interference. Again, phasing could be somewhat problematic here. Maybe there are enough homozygotes in the data set that the authors could at least confirm that some of the findings hold even for unphased data? Also, I'm concerned that unknown population structure could potentially play into these results. This would be difficult to test, obviously, given that it's unclear what models one should test specifically. However, one question that I think could be more easily answered is how likely it is to find such negative LD at other genomic loci. Are the resistance loci truly genomic outliers in this regard? I hope the authors can add some discussion about whether they think population structure may or may not provide a potential alternative explanation for the observed negative LD.

We agree that phasing is an issue if one infers evidence of these processes by just assessing whether or not we observed double resistant haplotypes. However, another main line of evidence is patterns of LD based on genotypes, rather than haplotypes (e.g. Figure 4A; using the plink1.9 algorithm). This analysis shows that based on genotypic associations, genotypes at ALS Trp-574-Leu are in negative LD with genotypes at ALS Ser-653-Asn (with an observed LD (*r*) value of -0.67 in Essex). Furthermore, it highlights repulsion between genotypes associated with either allele for nearly a 10 Mb stretch surrounding ALS. For these revisions, we compared the repulsion we see between these two ALS loci to similarly common (minor allele frequency > 0.20) and physically close (<500 bp apart) pairs of loci, finding that the repulsion we observe is more extreme than expected (*p* = 0.033). Beyond these pairwise comparisons, such a stretch of the repulsion across a 10 Mb segment is not observed (Figure 4—figure supplement 2).

While these regions are clear genomic outliers, we very much agree with the reviewer that the most likely explanation relates to a combination of local selective sweeps in separate populations followed by their spread. We have substantially revised the wording in the results (lines 386-389, 405-407) and discussion (lines 639-653) to make our inferences clearer, as well as added an analysis comparing local and genome-wide patterns of population structure through PCAs (Figure 4B). To reiterate, we believe that distinct spatial origins (and thus too some extent, population structure) contribute to the local structure we see around ALS. Strong selection on the haplotypes where these alleles arose before their contact has generated the signal of extended LD and repulsion between these local origins. However, given the co-occurrence of these alleles both globally (35% for ALS Trp-574-Leu and 19% for ALS Ser-653-Asn) and locally (within Essex, 29% and 44%, respectively), and even their co-segregation within 15 individuals, it is clear that these alleles have gone beyond their local origins and have been in contact for some time. It is clear that haplotype divergence in this region has been remarkably preserved, as individuals carrying different resistance mutations are not otherwise diverged (Figure 4B). Therefore, the evolution of target-site resistance seems to not only fit the first half of the Ralph and Coop (2010) model of spatial parallel origins (multiple origins and local increase), but also the latter half, the meeting and competing of these alleles. In particular, the breadth of the repulsion we see highlights that selection among resistant haplotypes must be occurring at this scale, through the competition of nearly 10 Mb-long haplotypes/genotypes associated with either ALS Trp-574-Leu or ALS Ser-653-Asn substitutions.

It is interesting to reflect on what may facilitate or prolong this haplotype competition: whether this may reflect inefficient selection and thus selective interference, whether the interaction of these alleles are neutral, or whether negative epistasis might drive selection against double resistant types. We agree this exercise is completely speculative, and so remove any conclusions about the underlying process and just offer these alternative scenarios in the discussion.

Reviewer #2 (Recommendations for the authors):I think this is a super interesting paper. It shows evidence for multiple origins (9) of drug resistance and also widespread migration (transmission) of resistance alleles between local populations. That in and of itself is worth publishing for me.I feel like what is missing from the paper is a discussion of what these results mean for preventing resistance in weeds. I think that it means that when we want to prevent resistance, we need to be concerned with preventing mutation (control pop size / selection locally) and we need to be concerned with gene flow between populations (do not share equipment and staff?). I think the paper could benefit from more translation of the results to a non-evolutionary genetics audience. I also would love to see more easy-to-digest information about the relevant herbicides and the weed itself. This way, you can make the paper more interesting for evolutionary geneticists who have never thought about herbicide resistance and you can introduce those who are interested in plants and agriculture to evolutionary genetics.

We appreciate the suggestion. We have added a paragraph to the conclusion (last paragraph, lines 729-743) and changed our introduction to provide more background and easy-to-digest information about *A. tuberculatus* and relevant herbicide use.

In my opinion, some major improvements could be made to the presentation. Part of that is eye for detail in figures (e.g., use same colors and same notation throughout) and text (e.g., line 118 why mention psbA here, but nowhere else? Line 126 why mention glyphosate here? These things make the text hard to follow.…).

We have gone through the text to remove any extraneous and distracting information. Thanks for pointing this out. We have also double-checked for colour consistency throughout the figures in the paper.

While we have clarified the first paragraph of the results you pointed out, we have kept our reference to psbA and glyphosate and clarified their context, as we initially assayed for all known TSR mutations (lines 153-159).

Some of the analysis on allele ages / evidence of recent selection should be presented differently before I am convinced.The same is true for the analysis on clonal interference. This should start with clarifying what is meant by interference / competition / etc.

We address these comments below in response to specific suggestions

Instead of TSR I would write "resistance allele" – I think that will help other readers.

TSR is the common language for these types of mutations in the weed science literature, however, to make it more readable to a wider audience, in many instances, we have changed TSR to resistance allele or lineage when appropriate. Thanks for the suggestion.

Figure 1: could Figure 1 be remade using the origins from Figure 2? Once you know that there are multiple origins, doing an analysis that ignores these origins doesn't make too much sense, I think. Can you use the same nomenclature as Table 1 and Figure 2?

We think it interesting to illustrate the regional sweep signals, as this reflects the influence that herbicide use has had on patterns of diversity and homozygosity at these regional scales. However, we take your point and have additionally provided sweep scan signals corresponding to these independent origins in Figure 2 – supplemental figure 5 and 6. In some cases these signals are relatively muted, perhaps reflecting both recombinational and mutational processes subsequent to their origins.

We have also made sure to use the same nomenclature for resistance mutations throughout (i.e. ALS Trp-574-Leu for ALS Trp-574-Leu etc).

Figure 2:1. First, use ARG to find origins. Conclusion: there are multiple origins!2. Then plot origins and fractions on a map. Conclusion: there is migration as well, though not panmixia (would it make sense to test that?).

To test whether the stratification we observe is more extreme than expected given the number of haplotypes mapping to each origin, we performed a permutation in which we randomized assignment of haplotypes across geographic regions (Ontario versus the Midwestern US), repeated 1,000 times, and recalculated the proportion of haplotypes of a given origin mapping to each geographic region (methods lines 824-831). This provides an expectation for the random or panmictic distribution of lineages corresponding to each origin given their frequency. Across the seven origins we characterize, we find only two that have more extreme stratification than expected under this null, further emphasizing the role of gene flow in the spread of resistance.

3. Finally test for recent selection. Here I would love to see some kind of strength of selection statistic rather than just p-value.

We agree that this would be very nice. However, this is not a feature of ARGweaver or Relate inherently. We had implemented another method (Stern, Wilton, and Nielsen 2019) to infer the strength of selection for one origin, and presented it in the previous version of this manuscript. However, when analyzing each origin, we find this approach yields highly unlikely results, and thus we believe our data must not be suitable for such a test. We have removed this analysis from our revised manuscript. We hope to be able to address the question of selection coefficients associated with herbicide resistance in the future, with a dataset more appropriate for such questions (i.e., randomly sampled with respect to herbicide resistance).

For one allele there is evidence for recent selection, but not selection since origin (right?) – ALS 574#4. Could you show a zoom in for the tree with the 0.02% cut-off for that allele?Conclusion of figure 2: multiple origins and gene flow are both important. For some alleles evidence that there has been selection since origin – suggests de novo evolution of resistance. But for some possibly standing genetic variation? I am not so convinced of that part of the analysis. Maybe it'd help to show haplotypes?

We agree that this would be very nice. However, this is not a feature of ARGweaver or Relate inherently. We had implemented another method (Stern, Wilton, and Nielsen 2019) to infer the strength of selection for one origin, and presented it in the previous version of this manuscript. However, when analyzing each origin, we find this approach yields highly unlikely results, and thus we believe our data must not be suitable for such a test. We have removed this analysis from our revised manuscript. We hope to be able to address the question of selection coefficients associated with herbicide resistance in the future, with a dataset more appropriate for such questions (i.e., randomly sampled with respect to herbicide resistance).

Suggestion: can you show haplotypes (like Harpak et al. on rats or Garud et al. on *Drosophila* or Williams and Pennings on HIV?)

One important addition to our ARG-based inference is including support for each monophyletic origin across MCMC samples (i.e., in Figure 2A). Based on our allele age rescaling, while we still find that one ALS Trp-574-Leu mutational lineage predates the onset of herbicide use, this lineage also shows relatively low support for monophyletic clustering across MCMC samples. Accordingly, we emphasize the lack of confidence we have in evidence for standing genetic variation for these alleles. In support of adaptation from de novo mutations, our estimates put all other ALS mutational lineages arising right around the onset of herbicide use suggesting that resistance alleles may only persist with recurrent selection from herbicides.

We have provided a visualization of the haplotypes corresponding to different ARG-inferred origins (Figure 2 – supplemental figure 4), along with figures showing how XPEHH and H12 change as we move across the chromosome for haplotypes mapping to each origin (Figure 2 supplemental figure 5 and 6).

Please label MidWest and Ontario. Could the pies have the same scale in the two regions?

We have labelled the Midwest and Ontario agricultural regions (Essex and Walpole). We have left the different pie size legends as we feel the different scales make a bigger impact on readability.

Suggestion: Use the same color and nomenclature scheme throughout. For example, the 653_7 origin should have that name and color throughout the figure and other figures.

We have double-checked that our colour scheme and nomenclature are consistent throughout.

Figure 3:The increase in pop size is clear and not surprising.The allele ages are interesting. Could they be plotted?

The allele ages are plotted in Figure 3b.

I am not convinced that there is evidence for SGV for 653#7, because its age is only about 30 years according to the figure. 3B and 2C are hard to reconcile in my head.

Again, these results have changed slightly with our revised results following rephasing and ARG inference with the most likely parameter set. We no longer see substantial evidence for adaptation from standing variation based on allele age estimates.

Figure 3B (the explicit allele age estimates) and Figure 2C (tree-based tests for selection) present the results from distinct tests and while they may be conceptually related, one is explicitly testing evidence for standing variation versus de novo mutation, while the other tests for non-neutral allele frequency change across a mutational lineage’s entire evolutionary history versus over just more recent timescales. In particular, our results highlight a scenario where these tests illustrated processes over different timescales – while we find evidence of de novo adaptation from allele age estimates, we also find that signatures of selection are stronger on timescales even more recent than the onset of herbicide use.

Additionally, it is reassuring to see consistent signatures of selection since a mutational lineage arose for the majority of mutations (i.e. Figure 2C), as this largely supports our inferences about adaptation on standing variation versus de novo mutation based on allele age estimates (i.e. Figure 3C), but without any assumptions about effective population size. Furthermore, the tree-based test illustrates heterogeneity in signatures of selection across origins, in that some origins show no sign of strong selection (ALS Trp-574-Leu #3, ALS Ser-653-Asn #3, PPO ΔGly210 #7) despite their inferred recent origins.

We have revised our discussion of these two approaches to emphasize our motivation for performing both tests (lines 303-310; Results), and how their interpretation differs (lines 609-620; Discussion).

Figure 4:This is about interactions between ALS574 and ALS 653. These are 300-ish bp apart.The resistance mutations never occur on the same haplotype which is surprising (how surprising? Could we get a prediction for how common this should have been if there was free recombination given the age of the alleles?).

We have a paragraph in the discussion which discusses the likelihood of finding a double resistant haplotype given the local recombination rate and their frequencies (lines 661-674).

Now, I'd say the next question is: how often do they occur in the same individual, given their commonality in each population. Like a HW-test. Is there selection against carrying two resistance alleles?

We have now provided the number of individuals that these alleles are found co-occurring within in the results = 16 / 152 sampled individuals (line 382). Furthermore, we find that our observed genotype frequencies are in clear violation of expectations under linkage equilibrium (line 345-347). However, this is not strong evidence for selection against carrying two resistance alleles, as we still don’t know whether recombination has had the chance to recombine these haplotypes on the same background based on their contact time, or what the fitness effects of the combined haplotype are compared to single mutations (discussed at lines 676-689).

I am not entirely convinced that there is evidence for competition. Or maybe it is not clear to me what the authors mean exactly by competition / interference.I am not sure my brain can follow Figure 4A – what would be the expectation here given multiple origins at both loci?

We have clarified what we mean by haplotype competition in the results at lines 386-389, 405-407 and in the discussion at lines 639-653 (additionally, in earlier responses in this document). We suspect that independent (spatially structured) origins are likely playing a key role in driving this signal. We point out that repulsion disequilibrium would be expected if these alleles were to have arisen on distinct backgrounds. Since their origins, however, these alleles have spread and met (both locally and globally), and given their independence from one another (e.g. lack of haplotypes harbouring both mutations, long-range repulsion), these resistant alleles along with their linked genetic variation across a 10 Mb haplotype, must be competing on their way towards fixation.

It is likely that multiple distinct origins of ALS Trp-574-Leu in Essex have slightly weakened our ability to detect signals of competition between mutations ALS Ser-653-Asn and ALS Trp-574-Leu (in fact it is possible that haplotypic competition is occurring between the unique origins of each mutation too). The larger number of origins of ALS Trp-574-Leu haplotypes mapping to Essex likely explains why individuals harboring ALS Trp-574-Leu are more spread out in our genotypic PCA compared to individuals harboring ALS Ser-653-Asn (in a 2Mb region around ALS, Figure 4B). However, given that one origin of ALS Trp-574-Leu does predominate within Essex (likewise for ALS Ser-653-Asn), it appears that we have enough power to detect a signal of haplotypic differentiation around the ALS locus regardless.

References:

Costea, Mihai, Susan E. Weaver, and François J. Tardif. 2005. “The Biology of Invasive Alien Plants in Canada. 3. Amaranthus Tuberculatus (Moq.) Sauer Var. Rudis (Sauer) Costea and Tardif.” Canadian Journal of Plant Science. Revue Canadienne de Phytotechnie 85 (2): 507–22.

Kreiner, Julia M., Darci Ann Giacomini, Felix Bemm, Bridgit Waithaka, Julian Regalado, Christa Lanz, Julia Hildebrandt, et al. 2019. “Multiple Modes of Convergent Adaptation in the Spread of Glyphosate-Resistant Amaranthus Tuberculatus.” Proceedings of the National Academy of Sciences of the United States of America 116 (42): 21076–84.

Neher, Richard A., and Boris I. Shraiman. 2009. “Competition between Recombination and Epistasis Can Cause a Transition from Allele to Genotype Selection.” Proceedings of the National Academy of Sciences of the United States of America 106 (16): 6866–71.

Rasmussen, Matthew D., Melissa J. Hubisz, Ilan Gronau, and Adam Siepel. 2014. “Genome-Wide Inference of Ancestral Recombination Graphs.” PLoS Genetics 10 (5): e1004342.

Speidel, Leo, Marie Forest, Sinan Shi, and Simon R. Myers. 2019. “A Method for Genome-Wide Genealogy Estimation for Thousands of Samples.” Nature Genetics 51 (9): 1321–29.

Stern, Aaron J., Peter R. Wilton, and Rasmus Nielsen. 2019. “An Approximate Full-Likelihood Method for Inferring Selection and Allele Frequency Trajectories from DNA Sequence Data.” PLoS Genetics 15 (9): e1008384.